# Gradient Information Matters in Policy Optimization by Back-propagating through Model

**Chongchong Li** [1]*, **Yue Wang** [2]†, **Wei Chen** [3]†, **Yuting Liu** [1], **Zhi-Ming Ma** [4] **& Tie-Yan Liu** [2]

[1] Beijing Jiaotong University
`{18118002,ytliu}@bjtu.edu.cn`
[2] Microsoft Research Asia
`{yuwang5,tyliu}@microsoft.com`
[3] Institute of Computing Technology, Chinese Academy of Sciences
`chenwei2022@ict.ac.cn`
[4] Academy of Mathematics and Systems Science, Chinese Academy of Sciences
`mazm@amt.ac.cn`

## Abstract

Model-based reinforcement learning provides an efficient mechanism to find the optimal policy by interacting with the learned environment. In addition to treating the learned environment like a black-box simulator, a more effective way to use the model is to exploit its differentiability. Such methods require the gradient information of the learned environment model when calculating the policy gradient. However, since the error of gradient is not considered in the model learning phase, there is no guarantee for the model's accuracy. To address this problem, we first analyze the convergence rate for the policy optimization methods when the policy gradient is calculated using the learned environment model. The theoretical results show that the model gradient error matters in the policy optimization phrase. Then we propose a two-model-based learning method to control the prediction error and the gradient error. We separate the different roles of these two models at the model learning phase and coordinate them at the policy optimization phase. After proposing the method, we introduce the directional derivative projection policy optimization (DDPPO) algorithm as a practical implementation to find the optimal policy. Finally, we empirically demonstrate the proposed algorithm has better sample efficiency when achieving a comparable or better performance on benchmark continuous control tasks. Codes are available at https://github.com/CCreal/ddppo

## 1 Introduction

Reinforcement learning (RL) is a powerful technique for solving the sequential decision making problems (Li, 2018; Sutton & Barto, 1998). Recent work on model-based RL (Nagabandi et al., 2018; Luo et al., 2018; Kurutach et al., 2018; Wang et al., 2019; Janner et al., 2019; Pan et al., 2020), has shown the power of first learning the environment model and then using it to do the policy optimization. Several methods are proposed to achieve the goal of getting similar performance by using fewer data, such as ensembles (Kurutach et al., 2018), probabilistic models (Chua et al., 2018), and meta-learning (Clavera et al., 2018).

In addition to treating the learned environment as a black-box simulator, a more effective way of using the model is to exploit its differentiability (Heess et al., 2015; Clavera et al., 2019; D' Oro & Jaśkowski, 2020; Amos et al., 2021), which is mainly focused in this paper. To get the policy updating direction, these methods compute the policy gradient directly by back-propagating through the model. Therefore, the model gradient is used in the calculation and its error will influence the accuracy of the policy gradient. However, since the traditional model learning only aims to get the accurate prediction for the next state and the reward, there is no guarantee for the accuracy of the

---

*This work was done when Chongchong Li was interning at MSRA.
†Corresponding author.

model gradient. In other words, the algorithm requires the accurate model gradient, but we only learn to decrease the prediction error which results in an objective mismatch.

In this paper, to address these problems, we first theoretically analyze the problem and then propose our solution based on the theoretical results. First of all, we present the convergence rate analysis for the policy optimization algorithms in which the policy gradient is calculated using the learned environment model. By taking the model gradient error into account, we can see that the gradients of the transition and reward models matter in the policy optimization. Specifically, the bias of the estimated policy gradient used to update the policy is not only introduced by the prediction error of the learned model but also introduced by the gradient error of the learned model. Furthermore, the policy gradient bias due to the different types of model error will finally influence the convergence rate of the policy optimization process.

Then, inspired by the theoretical results, we propose the two-model-based learning method. According to the policy gradient bias and convergence rate analysis, in order to optimize the policy efficiently and accurately, we need the learned environment model both with small prediction error and small gradient error. Therefore, we propose to set separate models for different purposes of usage. In the model learning phase, the prediction model aims to reduce the prediction error, and the gradient model focuses on minimizing gradient error. In the policy optimization phase, we will use the prediction model to rollout the data and use the gradient model to calculate the policy gradient.

To make the proposed method applicable, we introduce the directional derivative projection policy optimization (DDPPO) algorithm. Our first goal is to use data to estimate the gradient or Jacobian matrix for the environment model and use the estimator to learn the model's gradient explicitly. The challenge is that the state and action are usually with high dimensions and directly estimating the gradient or Jacobian matrix using data is intractable. Thus, we first estimate the directional derivative using data and then project the model's gradient or Jacobian matrix into these directions. Minimizing the error between the estimated directional derivative and the projection value, we can learn the model's gradient. Secondly, after learning the environment model with a more accurate gradient, we can leverage two-model-based learning method to do the policy optimization.

Finally, we conduct experiments on the simple environments and the benchmark MuJoCo continuous control environments. The experimental results verify our theoretical findings and demonstrate the effectiveness of the proposed method.

Our main contribution can be summarized as follows:

1. We theoretically depict how the different model errors influence the convergence rate of the model-based policy optimization algorithm. The result shows that the gradient error of the model indeed matters in the convergence of the policy optimization.

2. We propose the two-model-based learning method and the practical DDPPO algorithm which learns and uses two environment models (prediction model used for rollout and gradient model used to provide the gradient information) for the model-based policy optimization.

3. Empirically, we can achieve better sample efficiency in the MuJoCo continuous control tasks than state-of-the-art model-based and model-free methods .

## 2 PRELIMINARIES

**Reinforcement Learning:** We consider a discrete-time Markov decision process (MDP) $\mathcal{M}$, defined by tuple $(\mathcal{S}, \mathcal{A}, f, r, \gamma, p_0)$, where $\mathcal{S}$ and $\mathcal{A}$ are the state and action spaces, respectively. Here, $f : s_{t+1} = f(s_t, a_t, \epsilon_t)$ is the transition distribution, $r : s_t = r(s_t, a_t)$ is a reward function, $p_0$ is the initial state distribution and $\gamma$ represents the discount factor. We define the return $(J)$ as the expected sum of discounted rewards along a trajectory $(R = \sum_{t=0}^{\infty} \gamma^t r(s_t, a_t))$. The goal of reinforcement learning is to find a policy $\pi_\phi$ that maximizes the expected sum of discounted rewards, i.e.,

$$\max_\phi J(\phi) = \max_\phi \mathbb{E}_{\pi_\phi} R = \max_\phi \mathbb{E}_{\pi_\phi} \left[ \sum_{t=0}^{\infty} \gamma^t r(s_t, a_t) \right]. \tag{1}$$

Model-based RL is characterized by learning the transition model using data collected from interaction with the environment. Typically, we use a parametric function $\tilde{f}_\theta$ to denote the transition

model trained by a neural network. The state predicted by the learned model is defined as $\tilde{s}_{t+1}$ and $\tilde{s}_{t+1} \sim \tilde{f}_\theta$. Similarly, we use a parametric function $\tilde{r}_\theta$ to denote the reward model. Generally we use $M$ to denote the true model and $\tilde{M}$ the learned model. Then $\tilde{s}_{t+1}, \tilde{r}_t \sim \tilde{M}_\theta(s_t, a_t)$.

The model is trained by supervised learning, typically, via maximum likelihood:

$$J_{\tilde{M}^p}(\theta_p) = \mathbb{E}\left[\log \tilde{M}^p(s_{t+1}, r_t | s_t, a_t)\right], \tag{2}$$

where $(s_t, a_t, r_t, s_{t+1})$ are data collected from interaction with the environment.

**Policy Optimization According to the Policy Gradient Calculated Using Model:** After we learn the model, we can use the model to rollout trajectories and use the data to calculate the policy gradient for the purpose of policy optimization. Specifically, denote the policy parameters as $\phi$, we can calculate the return $J$ using generated data as $J_\pi(\phi) = \mathbb{E}\left[\sum_{t=0}^\infty \gamma^t \tilde{r}_t\right]$, where $\tilde{s}_{t+1}, \tilde{r}_t \sim \tilde{M}(\tilde{s}_t, \tilde{a}_t)$ and $\tilde{a}_t \sim \pi_\phi(\tilde{s}_t)$. To optimize the objective, one can use the traditional policy optimization methods such as A3C (Mnih et al., 2016), PPO (Schulman et al., 2017), SAC (Haarnoja et al., 2018). Beyond that, many previous works argue that we can use the model to calculate the policy gradient by back-propagating through the model (Heess et al., 2015; Clavera et al., 2019; Amos et al., 2021) leveraging the re-parameterization trick (Kingma & Welling, 2014).

Specifically, since we have learned the environment model, we can calculate the gradient of the objective w.r.t. $\phi$ by chain rule as shown in Equation 3 in which the gradients $\frac{dr_t}{da_t}, \frac{dr_k}{ds_k}, \frac{ds_k}{ds_{t+1}}, \frac{ds_{t+1}}{da_t}$ need to be calculated using the learned model.

$$\frac{dJ}{d\phi} = \mathbb{E}\sum_{t=0}^\infty \frac{dJ}{da_t}\frac{da_t}{d\phi} = \mathbb{E}\sum_{t=0}^\infty \left(\gamma^t \frac{dr_t}{da_t} + \sum_{k=t+1}^\infty \gamma^k \frac{dr_k}{ds_k}\frac{ds_k}{ds_{t+1}}\frac{ds_{t+1}}{da_t}\right)\frac{da_t}{d\phi}. \tag{3}$$

The expectation is taken over randomness from random policy and transition. To optimize the policy, we can update the policy parameters in the gradient direction. Since we don't have the true gradient, we use learned model to estimate the policy gradient denoted as $\frac{dR^{\hat{M}}}{d\phi}$. $\alpha_t$ is learning rate.

$$\phi_{t+1} = \phi_t + \alpha_t \frac{dR^{\hat{M}}}{d\phi}. \tag{4}$$

**Notations :** We use fraction notation $\frac{d\cdot}{d\cdot}$ to represent the gradient calculated by taking gradient on the true model and use the "hat" version $\frac{\widehat{d\cdot}}{d\cdot}$ to represent the gradient calculated on the learned model. We use the "tilde" notation $\tilde{\cdot}$ to represent the state, action or reward generated using the learned model and the without-tilde notation to represent those variables generated using the true model. For example, the notation $\frac{ds_{t+1}}{ds_t}$ denotes the gradient function value calculated by taking gradient for the output of the transition model $s'$ ($s' = f(s, a, \epsilon)$) with respect to its input $s$ given input value equals $s = s_t$. We use the notation $\epsilon_f$ to represent the transition model prediction error which means, for $\forall\, s, a$, we have $\mathbb{E}\left\|(\tilde{f}(s, a) - f(s, a))\right\| \leq \epsilon_f$. We use the notation $\epsilon_f^g$ to represent the gradient error of the transition model. For $\forall\, s, a$, we have

$$\mathbb{E}\left\|\frac{\widehat{ds'}}{ds} - \frac{ds'}{ds}\right\| + \mathbb{E}\left\|\frac{\widehat{ds'}}{da} - \frac{ds'}{da}\right\| = \mathbb{E}\left\|\frac{d\widehat{f}(s,a)}{ds} - \frac{df(s,a)}{ds}\right\| + \mathbb{E}\left\|\frac{d\widehat{f}(s,a)}{da} - \frac{df(s,a)}{da}\right\| \leq \epsilon_f^g. \tag{5}$$

Similarly, we use the notation $\epsilon_r^g$ to represent the gradient error of the reward model.

# 3 RELATED WORK

Due to the space limitation, we clarify several highly related works in this section and put the detailed related work into the Appendix A. Overall speaking, most of the model-based policy optimization algorithms aim to calculate the policy gradient accurately and efficiently. These works can be divided into two categories, i.e., how to learn the model and how to use the model.

**Learn the better model:** Chua et al. (2018) design an ensemble-based neural network to learn the true model. Zhang et al. (2021a) design the dropout mechanisms to improve the robustness of the

learned model. Zhang et al. (2020b) design the auto-regressive network structure to improve the accuracy of the model prediction.

**Consider the decision-making phase when learning the model:** These works are always called Decision-aware model learning (DAML). D'Oro et al. (2020) use policy gradient to get the re-weighted model prediction error for each sample. Farahmand et al. (2017) use the value function to get the re-weighted model prediction error for each sample.

**Use the learned model to argument data:** Janner et al. (2019) use the learned model to rollout the trajectories and use both the generated and the true trajectories to train the SAC algorithm. Yu et al. (2020); Lee et al. (2020) extend the similar idea to the offline model-based RL setting.

**To calculate the policy gradient by back-propagating through the model:** SVG (Heess et al., 2015) presents a framework for learning continuous control policies using back-propagating through the model. More recent works (Clavera et al., 2019; Amos et al., 2021) extend the idea by combining the learned Q-function and the SOTA policy optimization algorithms and thus achieve better performance. MAGE (D' Oro & Jaśkowski, 2020) computes the policy gradient using value gradient and proposes to reduce the value gradient through TD learning.

Different from the above methods, we further consider the influence of the model gradient error with respect to the convergence of the policy optimization. We will use multiple function approximators to separate different requirements for the model at the learning and using phase. Moreover, we can leverage the insight inside of these related works to better design our model-based algorithms.

## 4 MODEL-BASED POLICY OPTIMIZATION BY CONSIDERING THE MODEL GRADIENT ERROR

In this section, we will analyze the influence of the model error in the policy optimization process. First of all, for the policy optimization algorithms that using policy gradient calculated according to the learned model, we theoretically show that both the prediction error and the gradient error of the learned model matter for the policy gradient bias and the convergence rate. Secondly, based on the theoretical findings, we propose our method to learn the accurate model for policy optimization. Thirdly, we show the implementation details of our method and propose the DDPPO algorithm.

### 4.1 CONVERGENCE RATE FOR MODEL-BASED POLICY OPTIMIZATION

In this section, we will prove the finite sample bound for the model-based policy optimization algorithms. Due to the space limitation, we put the full proof into the Appendix C.

Due to the space limitation, we simply put the assumptions that is necessary for understanding the theoretical results here and put the detailed assumptions into Appendix C.

**Assumption 1.** *The transition model, the reward model and the policy function are all Lipschitz continuous. The state space and the reward are both bounded. For a fixed policy, the stationary distribution exists and has the uniform ergodicity property:*

$$d_{TV}\left(P(s_{t+\tau} \in \cdot | s_t = s), \mu_\theta(\cdot)\right) \le m\rho^\tau \ \forall \tau \ge 0, \ \forall s \in \mathcal{S}. \tag{6}$$

First of all, we analyze the bias of the policy gradient introduced by the model error. Recall the policy gradient formulation (Equation 3) in the section 2. In the model-based setting, we need to use the learned model to calculate the gradient value ($\frac{dr_t}{da_t}, \frac{dr_k}{ds_k}, \frac{ds_k}{ds_{t+1}}, \frac{ds_{t+1}}{da_t}$) in Equation 3. Since the model error is unavoidable, it is necessary for us to analyze the bias of the policy gradient introduced by the model error.

**Theorem 1.** *Suppose assumptions hold. The policy gradient bias $\epsilon_M$ can be bounded as*

$$\epsilon_M = \left\| \mathbb{E}\left(\frac{dR^{\widehat{M}}}{d\phi} - \frac{dR^M}{d\phi}\right) \right\| \le C_6 \epsilon_r^g + C_7 \epsilon_f^g + C_8 \epsilon_f, \tag{7}$$

*where $C_6, C_7, C_8$ are three constants related to the Lipschitz constants of the transition model and the reward model and $\gamma$.*

**Remark:** The above theorem shows that the difference between the policy gradient calculated using two different models can be upper bounded by the difference of the gradient of the reward model$\epsilon_r^g$,

the gradient of the transition model $\epsilon_f^g$, and the prediction of the transition model $\epsilon_f$. The result suggests that gradient error matters in estimating the policy gradient. Now, one natural conjecture is that the gradient error will influence the convergence of the policy optimization process. The next theorem depicts the influence rigorously.

Secondly, we analyze the convergence rate for the policy gradient algorithms. Since it is always a non-convex optimization problem, the commonly used measure is the speed of the gradient norm converges to zero. Recall the update rule (Equation 4). We update the policy parameters by following the estimated policy gradient direction. Intuitively, the model error will influence the convergence.

**Theorem 2.** *Suppose assumptions 3, 4, 5, 6,7, and 8 holds. $L_J$ is the Lipschitz constant of the objective function for the parameter $\phi$ and $L_s^g$ is the smoothness constant of the objective function for the parameter $\phi$. Suppose the policy parameters are updated as shown in Equation 4 and the learning rate is set as $\alpha_t$. We have*

$$\min_{0 < t < T} E \left\| \frac{dJ}{d\phi_t} \right\|^2 \leq \frac{2}{\sum_{t=1}^T \alpha_t} \left( 2U_J + 2\tau\alpha_0 L_J^2 + \sum_{t=1}^T \alpha_t^2 \left( L_J^2 {L_J^g}^2 \tau + L_J^g L_R^2 \right) \right) + 2L_J m \rho^\tau + \epsilon_M^2. \quad (8)$$

**Remark:** The above theorem shows the rate of the model-based policy gradient method converges to its stationary point. This setting is commonly used in the previous papers that analyze the convergence rate for the actor critic method with function approximation and non-convex optimization (Wu et al., 2020; Kumar et al., 2019; Qiu et al., 2021; Ghadimi & Lan, 2013). The results show that if we set the learning rate $\alpha_t$ that satisfies the condition $\lim_{T \to \infty} \frac{\sum_{t=1}^T \alpha_t^2}{\sum_{t=1}^T \alpha_t} = 0$, the policy optimization process will converge to near its stationary point with rate $\mathcal{O}\left( \frac{\sum_{t=1}^T \alpha_t^2}{\sum_{t=1}^T \alpha_t} \right)$. As an example, if we set the learning rate as $\alpha_t = \frac{1}{\sqrt{t}}$, we can get the convergence with rate $\mathcal{O}(\frac{1}{\sqrt{T}})$.

**Remark:** The bias is introduced by the policy gradient bias and the Markov noise. The bias introduced by the Markov noise can be arbitrarily small if we set $\tau$ sufficiently large which doesn't change the main order of the convergence rate. The bias introduced by the policy gradient bias is analyzed in Theorem 1. According to the theoretical results, to reduce policy gradient bias, we need to control the model prediction error and the model gradient error in the model learning phase.

### 4.2 Two-Model-Based Learning Policy Optimization

The theorem in the previous subsection indicates that both the model prediction error and the model gradient error are crucial for policy optimization. In this subsection, we will discuss how to learn and use the model taking both errors into account. Specifically, our approach repeatedly alternates between collecting samples from the environment, training the model, and optimizing the policy.

**Model Learning:** Different from the classical model-based methods, we design a two-model-based learning method to achieve the goal. We call these two models prediction models and the gradient models respectively. Our goal is to use the prediction model to predict the next state and the reward and to use the gradient model to calculate the policy gradient. Training of the prediction model is commonly through supervised learning, e.g., maximum likelihood with early stopping on a validation set (Janner et al., 2019; Clavera et al., 2019). Training of the gradient model is more complex. For the purpose of reducing the gradient error, we need to use the data collected from the environment to estimate the gradient of the environment model and construct the mean square gradient error for the parameter optimization.

**Policy Optimization:** To update the policy, we first generate trajectories by unrolling the latest policy on the prediction model. Then, we calculate the gradient of the objective by back-propagating through the gradient model. Finally, we update the parameters in the policy by following the policy gradient direction.

### 4.3 Directional Derivative Projection Policy Optimization: A Practical Implementation

In this subsection, we propose Directional Derivative Projection Policy Optimization (DDPPO) as an implementation of our proposed two-model-based learning method. The key question is how to use the data collected from the environment to estimate the gradient or Jacobian of the environment

---

**Algorithm 1** Directional Derivative Projection Policy Optimization

---

1: Initialize the policy $\pi_\phi$, predictive model $\tilde{M}^p_{\theta_p}$, gradient model $\tilde{M}^g_{\theta_g}$, value function $Q_\psi$
2: Initialize environment replay buffer $\mathcal{D}_{env}$, model replay buffer $\mathcal{D}_{model}$
3: **repeat**
4:     Take an action in real environment with policy $\pi_\phi$, and add transition $(s, a, s')$ to $\mathcal{D}_{env}$
5:     Train predictive model $\tilde{M}^p_{\theta_p}$ and update $\theta_p$ by maximizing $J_{\tilde{M}^p}(\theta_p)$
6:     Train gradient model $\tilde{M}^g_{\theta_g}$ and update $\theta_g$ by maximizing $J_{\tilde{M}^g}(\theta_g)$
7:     **for** $N_{rollouts}$ steps **do**
8:         Sample $s_t$ uniformly from $\mathcal{D}_{env}$
9:         Perform $k$-step model rollout on $\tilde{M}^p_{\theta_p}$ starting from $s_t$ with policy $\pi_\phi$
10:        Add samples to $\mathcal{D}_{model}$
11:     **end for**
12:     $\mathcal{D} \leftarrow \mathcal{D}_{env} \cup \mathcal{D}_{model}$
13:     **for** $N_{update\,Q}$ steps **do**
14:         Update $\psi$ using data form $\mathcal{D}$: $\psi \leftarrow \psi - \lambda_Q \nabla_\psi J_Q(\psi)$
15:     **end for**
16:     **for** $N_{update\,Policy}$ steps **do**
17:         Sample trajectories of length $H$ with policy $\pi$ and predictive model $\tilde{M}^p_{\theta_p}$
18:         Update $\phi$ using sampled trajectories and gradient model $\tilde{M}^g_{\theta_g}$: $\phi \leftarrow \phi + \lambda_\pi \nabla_\pi J_\pi(\phi)$
19:     **end for**
20: **until** The policy performs well in real environment $M$
21: **return** Optimal policy $\pi_{\phi^*}$

---

model. Note that the dimension of the Jacobian of the learned environment model is the product of the state's dimension and the action's dimension, which is usually large. Directly estimate the gradient or Jacobian matrix is intractable. For model-based RL, we cannot estimate gradients by perturbing one dimension of a state or action and estimate the partial derivative, because we can only sample data from the environment, and this is contrary to the purpose of efficiency. Besides these, we will also discuss how to design two models to rollout and calculate gradients respectively when optimizing the policy. The overall algorithm pseudo-code is shown in Algorithm 1. More details for explaining the pseudo-code can be found in Appendix B.1.

**Model Learning: Prediction loss and the gradient loss.**

The prediction model $\tilde{M}^p$ is trained following Janner et al. (2019) via maximum likelihood (Equation 2) To improve the ability of models to portray complex environment, we use a bootstrap ensemble of models $\{\tilde{M}_{\theta_1}, \dots, \tilde{M}_{\theta_B}\}$ which is consistent with Janner et al. (2019); Clavera et al. (2019).

Different from the prediction model, the gradient model $\tilde{M}^g$ aims to better characterize the gradient of the environment instead of an accurate prediction. Hence, we hope to trade-off the original maximum likelihood objective by a gradient loss term that constraints the Jacobian of the learned model. Although the model's gradient or Jacobian matrix is difficult to estimate directly on account of the high dimensions, the directional derivative can be estimated by finite difference using data sampled. Meanwhile, the directional derivative of the learned model can be calculated by projecting the model's gradient or Jacobian matrix into a certain direction.

Hence in this paper, we use the error between the estimated directional derivative and the projection value to constrain the learned model's gradient. Note that for an input $\boldsymbol{x} = (s, a)$, and the corresponding output $M(\boldsymbol{x})$ with dimension $d$, the projection of the Jacobian along a vector $v$, i.e., $\nabla_{\boldsymbol{x}} M(x) \cdot \frac{\boldsymbol{v}}{||\boldsymbol{v}||}$, can be estimated by finite difference, $\frac{M(\boldsymbol{x}+\boldsymbol{v})-M(\boldsymbol{x})}{||\boldsymbol{v}||}$. Therefore, we use the finite difference as a target to constrain the Jacobian of the gradient model. Since the finite difference converges to the directional derivative when the norm of $\boldsymbol{v}$ goes to zero, the data point $\boldsymbol{x} + \boldsymbol{v}$ should be close to $\boldsymbol{x}$. So in this paper, we sample the nearest $n$ data points of $x$ to calculate $n$ directional derivatives in $n$ different directions and use the finite difference to constrain the Jacobian of the learned model. Sampling the nearest points is for a more accurate estimation of the directional derivatives since the finite difference is closer to the true directional derivative if the two inputs are closer. Then, the gradient loss item can be constructed as follows:

$$R = \frac{1}{n} \sum_{i=1}^{n} \frac{1}{d} ||\nabla_{\boldsymbol{x}} \tilde{M}(\boldsymbol{x}) \cdot \boldsymbol{v}_i - (M(\boldsymbol{x}_i) - M(\boldsymbol{x}))||^2, \tag{9}$$

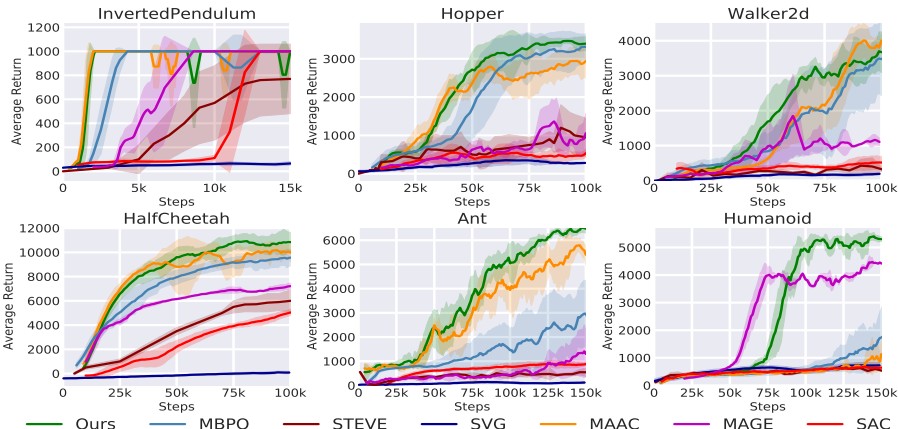

Figure 1: Learning curves of our algorithm and baselines on six different MuJoCo environments. Solid curves express the mean of five trials. Shaded regions correspond to standard deviation among five trials. Our method has better sample efficiency than the model-based baselines. The improvement of performance over previous algorithms is large on complex environments, i.e., Ant and Humanoid.

where $v_i$ denotes directions $x_i - x$, and $x_i$ are the nearest $n$ points of $x$ in the buffer. Intuitively, the gradient at $x$ can be constrained if there are enough points close enough to $x$. And a relatively small gradient loss item means the learned model should have a gradient such that its first-order approximation is consistent with the given data. By leveraging the gradient loss term, the gradient mode $\tilde{M}^g$, is trained via a multi-objective loss:

$$J_{\tilde{M}^g}(\theta_g) = \mathbb{E}\left[\log \tilde{M}^g(s_{t+1}, r_t | s_t, a_t)\right] + w \cdot R. \tag{10}$$

Note that here we don't prefer to learn gradient model using gradient loss term only. We can just use the data in the buffer to estimate a few of the directional derivatives since the points sampled should be close to the $x$ to make the estimation correct. So here we also use the prediction loss to constrain the hypothesis space of the learned model to prevent large bias local optimal. Finally, the trained gradient model and the prediction model will be used for policy optimization.

**Model Usage: Policy optimization by back-propagating through model.** In our DDPPO algorithm, to train the policy, the objective function we use is constructed by following SAC (Haarnoja et al., 2018), since SAC can achieve outstanding performance. The policy is learned by rolling out the prediction model for $H$ steps and maximizing the following objective function:

$$J_\pi(\phi) = \mathbb{E}\left[\sum_{t=0}^{H-1} \gamma^t(\tilde{r}_t - \alpha \log \pi_\phi(a_t | \tilde{s}_t)) + \gamma^H(\tilde{Q}(\tilde{s}_H, a_H) - \alpha \log \pi_\phi(a_H | \tilde{s}_H))\right], \tag{11}$$

where $\tilde{s}_{t+1} \sim \tilde{M}^p_{\psi_p}$, $\tilde{s}_0$ random sampled from the environment buffer $\mathcal{D}_{env}$, $a \sim \pi_\phi$, $\alpha$ is the temperature hyperparameter in SAC. Note that the policy can be learned by computing the derivative of the objective since the learned prediction model $\tilde{M}^p$ is differentiable. However, in our approach, we replace the gradient $\nabla_{\tilde{s}_t, a_t} \tilde{M}^p(\tilde{s}_t, a_t)$ by $\nabla_{\tilde{s}_t, a_t} \tilde{M}^g(\tilde{s}_t, a_t)$ to calculate the full gradient, believing in that the gradient model can capture the true gradient better than the prediction model.

As for the $Q$-value function, it can be trained by minimizing the Bellman residual. In practice, we follow SAC (Haarnoja et al., 2018) that trains two $Q$-functions to mitigate positive bias which is known to degrade the performance of value-based methods (Hasselt, 2010; Fujimoto et al., 2018), and make use of target $Q$-function which has been shown to stabilize training (Mnih et al., 2015).

## 5 EXPERIMENT RESULTS

The goal of our experimental evaluation is to answer the following questions: (1) How does our algorithm perform on benchmark reinforcement learning tasks, compared to state-of-the-art model-based and model-free algorithms? (2) Does it matter to explicitly learn the gradient of the model by using the gradient loss? (3) How does the sensitivity of our algorithm when changing the hyperparameters $n$ and $w$ that are related to the gradient model learning? (4) Why build two models? To answer these questions, we evaluate our approach on model-based continuous control benchmarks in the MuJoCo (Todorov et al., 2012). Experimental details can be found in Appendix B.2.

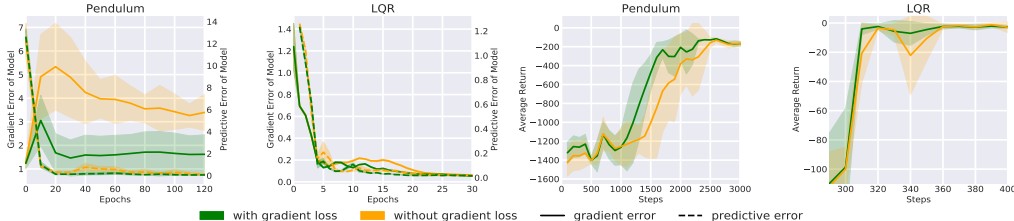

Figure 2: The left two figures show learning curves of models with and without gradient loss on two simple environments. Data used to train each model is collected by a random policy. Solid curves express the mean of gradient error of five trials. Dotted lines express the mean of prediction error during the training of the model. Shaded regions correspond to standard deviation among five trials. The model with gradient loss has a lower gradient error than the model without gradient loss. The improvement is more pronounced in nonlinear cases, i.e., Pendulum environment. The right two figures show the overall performance of using two different types of model, which yields the importance of considering the gradient loss.

## 5.1 COMPARISON WITH STATE-OF-THE-ARTS

To answer the first question, we compare our method against state-of-the-art model-based and model-free methods in terms of sample complexity and asymptotic performance. Model-free baseline includes SAC (Haarnoja et al., 2018), which is a widely accepted baseline. For model-based methods, we select MBPO (Janner et al., 2019) and STEVE (Buckman et al., 2019) as our baseline which both use short-horizon model-based rollouts. We also compare to model-based methods SVG (Heess et al., 2015), which firstly uses the derivative of models to learn the policy, and MAAC, (Clavera et al., 2019) similar to SVG, however entirely relies on the predictions of the model, removing the need for likelihood ratio terms. MAGE (D' Oro & Jaśkowski, 2020) is added to comparison since this method focuses on the gradient of the $Q$-function.

The learning curves for all methods, shown in Figure 1, highlight the strength of our method in terms of performance and sample complexity. In all the MuJoCo simulator environments, especially in two higher-dimensional tasks, Ant and Humanoid, our method learns faster and has better efficiency than previous model-based and model-free methods. Note that MAGE learns fastest on Humanoid, and the trick of this algorithm can also be applied to our method. However, even without the help of tricks in MAGE, we can still achieve better asymptotic properties. We get a return around 5400 in the Humanoid environment using just over 100 thousand interactions with the environment, while around three times as many interactions are needed for MBPO to achieve the same performance.

## 5.2 GRADIENT ERROR ANALYSIS

In this subsection, we answer the second question, i.e., does it matter to explicitly learn the gradient of the model by using the gradient loss? Firstly, we aim to analyze the difference between the gradient of the true model and the learned model. We collect experience using a random policy and train the model with and without gradient loss respectively. Then we calculate the MSE error of the gradient of the model. Figure 2 shows how the gradient error and the prediction error changes during training of the model on two simple environments that the true gradient is analytical available. From the results in Figure 2 we can see that the prediction error for both models decreases to zero while the gradient error of the model without gradient loss keeps larger than that of the model with gradient loss. The learning curves highlight the strength of adding a gradient loss to the model in terms of high prediction accuracy and accurate gradient estimation under the simple case.

## 5.3 ABLATION STUDY

To answer the third question, i.e., how does the sensitivity of our algorithm when changing the hyper-parameters, and to show the results in a comprehensive way, we use a low-dimensional environment, Hopper, and a high-dimensional environment, Ant, as our representative and then conduct two sets of experiments in them. Figure 3 shows the results. The left column shows how the efficiency of our algorithm changes by tuning $n$. The results demonstrate that, regardless of the value $n$, our method is better than our variant without gradient loss. The middle column shows the results for tuning $w$. The results indicate that the weight is recommended to be smaller for a higher-dimensional environment and the performance is not sensitive to the number of directional derivatives used to construct the gradient loss, even only one directional derivative used can also help to improve the performance.

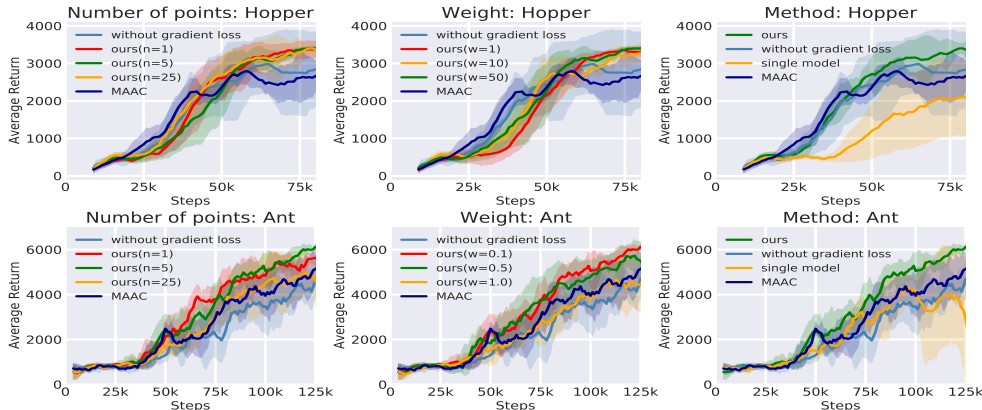

Figure 3: **Left:** This figure indicates our method is not sensitive to the number of directional derivatives used to construct the gradient loss. **Middle:** The performances of our method with different $w$, the coefficient in front of the gradient loss. The weight is recommended to be smaller for a higher-dimensional environment. **Right:** Our method outperforms the variant (without gradient loss) which underpins the importance of considering the gradient error when training the model. And the higher performances than the variant (single model) that only uses one model highlight the strength of the two-model-based structure.

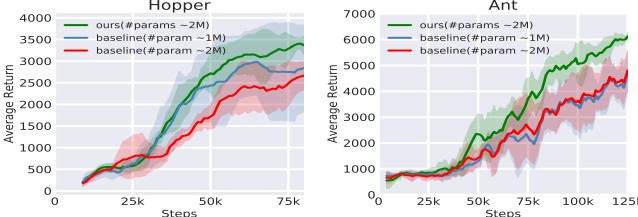

Figure 4: Our method performs better than baselines with a similar number of parameters which supports that the improvement is not actually from the additional parameters. Results for all six domains are shown in Figure 7 in Appendix B.3.

Now we answer the last question, i.e., why build two models. To investigate the importance of the gradient loss and role of two-model-based, we remove the gradient loss in our method and retrain the policy (without gradient loss) to see the importance of considering the gradient loss. To see the impact of training two models for prediction and gradient calculation respectively, we design a variant of our method that trains only one model with gradient loss used for both purposes (single model). The experimental results are shown in the right column of Figure 3. The results underpin the importance of considering the gradient error when training the model if back-propagating through it. On the other hand, the results highlight the strength of the two-model-based structure. We also draw the prediction error of both models in our algorithm and the results are shown in Appendix B.3. The prediction error of the prediction model is lower than that of the gradient model, which further validates the idea of two-model-based learning. To address the concern that the improvement of the performance may be caused by more parameters, we design a variant of our model that trains a model with double parameters, however without gradient loss. The experimental results are shown in Figure 4 that yields the importance of considering gradient information.

## 6 CONCLUSIONS AND FUTURE WORK

In this paper, we theoretically analyze how the model gradient error and the model prediction error influence the policy gradient bias and the convergence rate of model-based policy optimization. The theoretical results motivate us to design a two-model-based learning method for policy optimization. We design the learning and using methods for the two models respectively by taking the model gradient information into account and propose DDPPO algorithms as a practical implementation. Our algorithm can achieve superior sample efficiency than state-of-the-art model-based reinforcement learning algorithms on challenging high-dimensional continuous control tasks. For future work, it is enticing to design a better gradient estimator in order to optimize the gradient model more efficiently. The idea about learning the model according to its usage instead of only considering its prediction error is also general enough to be applied to other problems beyond reinforcement learning.

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

## A    RELATED WORK

Reinforcement learning (RL) is a powerful technique for solving the sequential decision making problems (Li, 2018; Sutton & Barto, 1998). The sequential decision-making problem is always formulated as the Markov decision process (MDP) framework. The agent uses policy to determine its action according to the current state of the environment, interacts with the environment, and gets the immediate reward from the environment as feedback. Then, the environment changes to the next state. The goal for reinforcement learning is to find the optimal policy that can achieve the highest expectation of the summation of the future reward. Recent work on model-based RL (Nagabandi et al., 2018; Luo et al., 2018; Kurutach et al., 2018; Wang et al., 2019; Janner et al., 2019; Pan et al., 2020), has shown the power of first learning the environment model and then use it to do the policy optimization. These methods all achieve similar performance by using fewer data compared to their model-free counterparts.

**Model-free and Model-based Methods:** Model-free approaches, as a general-purpose tool for learning complex policies (Mnih et al., 2015; Lillicrap et al., 2016; Haarnoja et al., 2018), has the problem of low-efficiency (Janner et al., 2019), which limits the application in real-world physical systems where data collection can be an arduous process. Model-based Reinforcement Learning attempts to reduce sample complexity while maintaining the asymptotic performance. The learned model can be viewed as a black-box simulator and then used for training a model-free policy (Nagabandi et al., 2018; Luo et al., 2018; Kurutach et al., 2018; Wang et al., 2019; Janner et al., 2019; Pan et al., 2020). Tools like ensembles (Kurutach et al., 2018), probabilistic models (Chua et al., 2018), and meta-learning (Clavera et al., 2018) are widely used in model-based RL. A major problem of model-based RL is model-bias (Deisenroth & Rasmussen, 2011) caused by a compounding error effect of long-term predictions. To address this problem, a popular approach is to use interpolation between different horizon predictions (Buckman et al., 2019; Janner et al., 2019) and interpolating between model and real data (Kalweit & Boedecker, 2017). Different from the traditional model based method, we argue to learn the model beyond the accurate prediction and to use the model more efficiently by calculating the policy gradient analytically through the learned model.

**Improved Model-based Methods:** Several issues have been studied to achieve better performance in model-based RL. Malik et al. (2019) explore which uncertainties are needed for model-based RL. Further research implements a masking mechanism based on the model's uncertainty estimation to decide whether the model should be used or not (Pan et al., 2020). Dropout mechanisms are also applied to model-based RL algorithms to improve the robustness while maintaining high sampling efficiency (Zhang et al., 2021b). Efficient exploration is also considered in model-based RL (Curi et al., 2020; Song & Sun, 2021; Yao et al., 2021). For model learning, Ke et al. (2018) use latent variables to carry future information that improves long-term predictions, Tomar et al. (2021) use model-invariance state abstraction to improve generalization. Kégl et al. (2021) firstly design metrics to evaluate the various popular generative models when using them on the control problem. Zhang et al. (2021a) shows that tuning of hyperparameters dynamically improves the performance compared to using static hyperparameters which are kept fix for the whole training. This operation can be seen as a simple treatment of model uncertainty. Considering that we also need the prediction model at the model learning and using phase, we can leverage the useful ideas inside of these methods to help us to build a robust prediction model and use it efficiently.

**Decision-aware Model Learning:** Decision-aware model learning (DAML) (Lovatto et al., 2020; Nair et al., 2020; Abachi et al., 2021; D'Oro et al., 2020; Farahmand et al., 2017; Modhe et al., 2021) considers the problem that how we can learn the model that can be better leveraged by the policy optimization or Q learning. D'Oro et al. (2020) using policy gradient to re-weight the model prediction error for each sample. Farahmand et al. (2017) use the value function to re-weight the model prediction error for each sample. Please note that, although the main goal about learning the model by considering the follow-up decision making problems seems very similar, our ideas and methods are totally different from DAML. DAML is still the traditional rollout-based model-based method with a re-weighted prediction loss.

**Differentiable Model-based Method:** A more effective way of using the model is to exploit its differentiability. SVG (Heess et al., 2015) presents a framework for learning continuous control policies using back-propagating through the model. However, SVG just uses real trajectories leveraging likelihood ratio term which in turn increases the variance of the gradient estimate. Recently,

Amos et al. (2021) extend SVG further leveraging the soft update (Haarnoja et al., 2018) and model-based value expansion (Feinberg et al., 2018). Different from SVG, MAAC (Clavera et al., 2019) entirely relies on the predictions of the model, removing the need for likelihood ratio terms, and achieves better performance than MBPO leveraging model-based value expansion for updating policy and Q-function. The plan horizon in these methods allows to trade-off between the accuracy of the learned model and the accuracy of the learned Q-function. In addition to back-propagating through the path, MAGE (D' Oro & Jaśkowski, 2020) uses the differentiability of the learned model to compute gradient targets in temporal difference learning. Our method is different from these methods. Different from these methods, our main contribution is that we propose to use two models to learn the model prediction and the model gradient respectively. By explicitly construct the gradient estimator, our gradient model can produce a more accurate policy gradient and thus benefits for the policy optimization.

**Outside of RL:** Several algorithms outside of RL adopt gradient penalties into the loss function when training a neural network, known as double backpropagation (Drucker & Le Cun, 1992). Sobolev training (Czarnecki et al., 2017) is most related to this topic, which tries to learn both the value and gradient of a target function during supervised training. However, Sobolev training requires the ground-truth gradient available. In the model-based reinforcement learning setting, we do not know the ground-truth gradient.

# B  ALGORITHM DETAILS

## B.1  IMPLEMENTATION OF DDPPO

### B.1.1  DIRECTIONAL DERIVATIVE

The directional derivative of a function $f$ with respect to a vector $v$ at a point (e.g., position) $x$ may be denoted by $\nabla_v f(x)$ that is a limit: $\nabla_v f(x) = \lim_{h \to 0} \frac{f(x+hv)-f(x)}{h}$. If the function $f$ is differentiable at $x$ then the directional derivative exists along any vector $v$ and we have: $\nabla_v f(x) = \nabla f(x) \cdot \frac{v}{|v|}$. Note that if $f$ is a scalar function and the dimension of $x$ is high, then $\nabla f(x)$ is a vector and difficult to estimate, however $\nabla_v f(x)$ is a scalar, and easy to estimate by finite-difference. This intuition helps us to design the constraint on the gradient of the learned model using data samples.

### B.1.2  BACK-PROPAGATING THROUGH GRADIENT MODEL

In our method, we use the prediction model to rollout, however use the gradient model to calculate the gradient when performing policy optimization. This procedure is like SVG (Heess et al., 2015) that gets predictions using the true environment however gets the gradient using a learned model. Note that reparameterization and important sampling weights are needed for this method. In our paper, we use the learned model as a deterministic model that only uses the mean outputted as the prediction. Now we show an example of calculating the gradient. When predicting, we first receive a state $s$ and use the policy to get action $a$, then using the prediction model to get the next state and reward $s', r$. Note that we need to memory the computation graph for getting the action. Then we use the gradient model to calculate the gradient by feeding the $(s, a)$ to it and back-propagation. Combined with the memorized computing graph we can get the final policy gradient.

### B.1.3  IMPLEMENTATION

Our algorithm can be divided into three main parts: model learning, policy optimization, and Q-function learning. And our approach alternates between collecting samples from the environment, training the model, updating Q-function, and policy optimization. The overall algorithm pseudo-code is demonstrated in Algorithm 1.

Firstly, we collect data using the current policy and add it into the environment replay buffer $\mathcal{D}_{env}$. Then the prediction model is trained using data from $\mathcal{D}_{env}$. The gradient model is also trained with the multi-objective loss. The third step is to perform $k$ step model rollout on the prediction model $\tilde{M}_{\theta_p}^p$ with the current policy, where $k$ is increased over time which proposed by Janner et al. (2019) to achieve better performance. Those samples are appended to the replay buffer $\mathcal{D}_{model}$, together with samples from $\mathcal{D}_{env}$, used to update the $Q$-function. To update the policy, firstly, we obtain

Table 1: Hyperparameter settings for DDPPO results shown in Figure 1.

| Environment Name | InvertedPendulum | Hopper | Walker2D | HalfCheetah | Ant | Humanoid |
|---|---|---|---|---|---|---|
| epochs | 15 | 100 | 100 | 100 | 150 | 150 |
| environment steps /epoch | 1000 | | | | | |
| ensemble size | 7 | | | | | |
| $G_1$ /environment step | 10 | | | | | |
| $G_2$ /environment step | 10 | | | | | |
| $H$ | 3 | 2 | 3 | | | |
| $n$ | 10 | 25 | 25 | 5 | | |
| $w$ | 10 | 50 | 0.1 | 1.0 | 0.1 | 0.1 |

imaginary trajectories by unrolling the latest policy on the prediction model $\tilde{M}^p_{\theta_p}$ for $H$ steps from a randomly sampled state in $\mathcal{D}_{env}$. Then we update policy by calculating the gradient of the objective in equation 11, while using the gradient model $\tilde{M}^g_{\theta_g}$ instead of $\tilde{M}^p_{\theta_P}$.

In our approach, we apply the soft Q-update (Haarnoja et al., 2018) which minimizes the following objective:

$$J_Q = \mathbb{E}\left[\left(\tilde{Q}(s_t, a_t) - \left(r(s_t, a_t) + \gamma\tilde{Q}(s_{t+1}, a_{t+1}) - \alpha\log(\pi_\phi(a_{t+1}|s_{t+1}))\right)\right)^2\right]. \tag{12}$$

The temperature hyperparameter is automatically tuned by adjusting the expected entropy over the visited states to match a target value (Haarnoja et al., 2018) with the following objective:

$$J_\alpha = \mathbb{E}\left[-\alpha\log\pi_\phi(a_t, s_t) - \alpha\bar{\mathcal{H}}\right], \tag{13}$$

where $\bar{\mathcal{H}}$ is the target entropy.

## B.2 EXPERIMENT SETTINGS

### B.2.1 ENVIRONMENT SETTINGS

We evaluate our approach on six continuous control benchmark tasks in the MuJoCo (Todorov et al., 2012) simulator in our experiments: InvertedPendulum-v2, Hopper-v2, Walker2d-v2, HalfCheetah-v2, Ant-v2 and Humanoid-v2. Standard full-length versions of these tasks are used. Note that the Ant and Humanoid environments are truncated observations which is consistent with MBPO (Janner et al., 2019). The code for building Ant and Humanoid environment is provided by Janner et al. (2019).[1]

### B.2.2 HYPERPARAMETER SETTINGS

Table 1 shows the hyperparameters used for DDPPO to achieve results shown in Figure 1. $G_1$ is the number of updates for policy per environment step, $G_2$ is the number of updates for critic per environment step. $H$ is the rollout length for updating policy. $n$ is the number of directional derivatives used to construct the gradient loss. $w$ is the coefficient in front of the gradient loss when updating the gradient model. Note that $H$ is used to trade-off between the accuracy of the learned model and the accuracy of the learned Q-function. As shown in the right figure, the results by setting a small $H$ is close to the results of MBPO, while a larger $H$ may cause worse results. We try different steps and propose 2 for Hopper and 3 for other environments. We employ 5000 (200 for the InvertedPendulum environments) warmup steps of interaction with the environment before starting to update the actor and the critic. The network structure of both the prediction model and the gradient model is the same

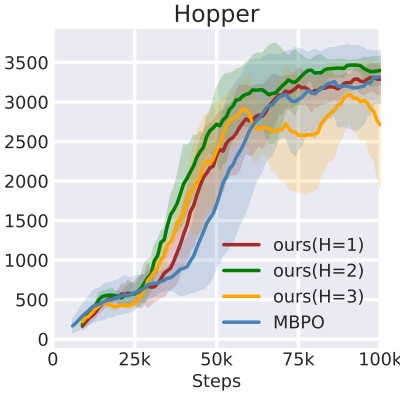

[1]https://github.com/JannerM/mbpo/tree/master/mbpo/env

as MBPO. Both models are trained after every 250 steps of interaction with the environment. Every time after model training, we perform rollout for $k$ steps and add the collected data to the buffer used for updating the critic. $k$ is increased with the number of interactions. For InvertedPendulum, Ant, HalfCheetah, and Walker2D $k$ keeps fixed. For Hopper $k$ increases from 1 to 15 between epoch 25 to epoch 155. For Humanoid $k$ increases from 1 to 25 between epoch 25 to epoch 155. The policy and $Q$-value function are updated 10 times per environment step which is less than the number of MBPO. For updating the policy we rollout on the prediction model for $H$ steps with batch size 256.

### B.2.3    EXPERIMENT DETAILS FOR BASELINES

For MBPO, we directly use the reported number given by Janner et al. (2019)[2]; For SAC we use the codes and hyperparameters available[3]; For MAGE, we use the codes available[4] and hyperparameters from the author. For MAAC, the model structure and hyperparameters such as learning rate are the same as MBPO, and the extra hyperparameter, $H$, we try different steps and set 2 for hopper and 3 for others which is consistent with ours. For other baselines, we use the codes or results from Wang et al. (2019)[5]. Note that the number of layers and the number of nodes of Q value functions are all the same for SAC, MBPO, MAAC, and ours. The number of layers and the number of nodes of the learned model are all the same for MBPO, MAAC, and ours. Finally, we keep unimportant hyperparameters of our method as consistent as possible to SAC or MBPO to keep it as fair as possible.

### B.3    MORE RESULTS FOR ABLATIONS

The results in Figure 3 show the importance of considering the gradient error when training the model if back-propagating through it and highlight the designed two-model-based learning procedure. We extend the experiments in Section 5.3 to more environments. The results in Figure 5 show the performance of our method compared to the variants of our method. The results in Figure 6 show that the prediction models in our algorithm have lower prediction error than the gradient models, which confirm our conjecture that the prediction and gradient calculation should be done by the corresponding model.

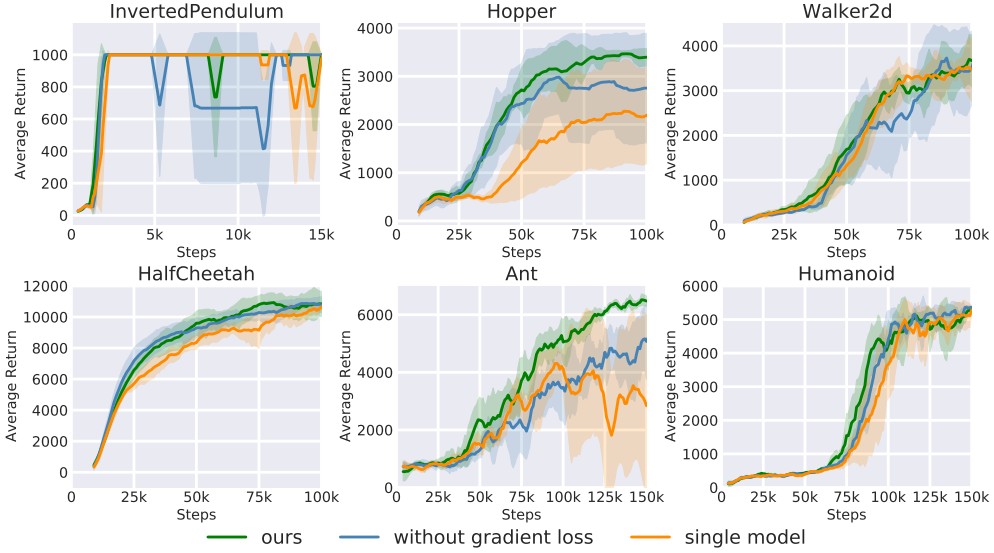

Figure 5: The impact of adding the gradient loss and the importance of two-model-based learning.

---

[2]https://github.com/JannerM/mbpo

[3]https://github.com/pranz24/pytorch-soft-actor-critic

[4]https://github.com/nnaisense/MAGE

[5]https://github.com/WilsonWangTHU/mbbl

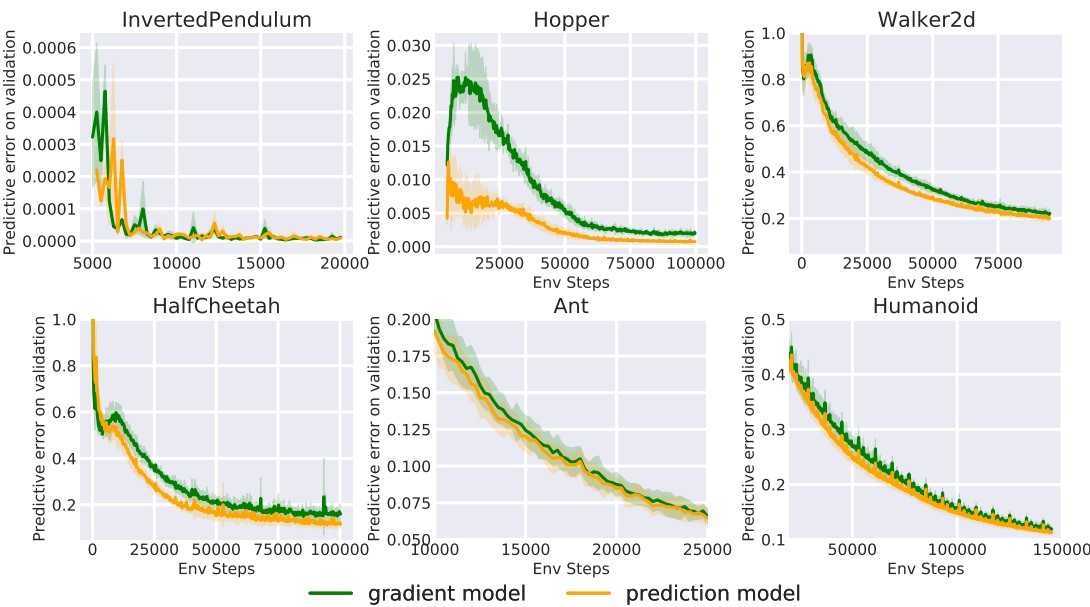

Figure 6: Prediction error of the prediction model and the gradient model on validation during the training of the agent. Directly adding the gradient loss item to the model training loss may influence the learning for prediction. This may be the reason why the performance of our variants (single model) is not even as good as using the model without consideration of the gradient loss for some environments, such as Hopper, HalfCheetah, and Humanoid.

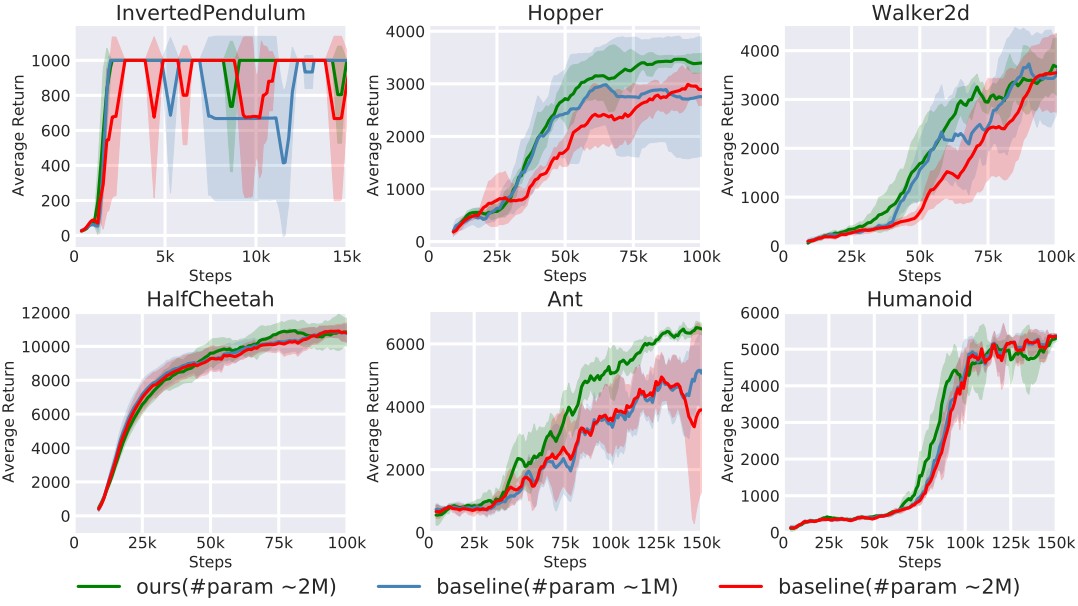

Figure 7: The higher performances than the variant with a similar number of parameters support that the improvement of our method is not actually from the additional capacity.

## C    PROOFS OF THE THEOREM

In this section, we present the proof of the main theorems in section 3.

First of all, we summarize the necessary assumptions here. These assumptions are commonly used for analyzing reinforcement learning algorithms.

**Assumption 2** (Stationary). *For $\forall t_1 < t_2$ and $s_{t_1}^0, s_{t_1}^1$, denote $\tau = t_2 - t_1$ there exist $L_1(\tau) < \infty$, $\tau = t_2 - t_1$, such that $\left\| \mathbb{E} \left( \frac{s_{t_2}^1 - s_{t_2}^0}{s_{t_1}^1 - s_{t_1}^0} \right) \right\| \leq L_1(\tau)$. And $L_1(\tau)$ can be uniformly upper bounded by $L_1 < \infty$. Here, $\| \mathbb{E} \left( \frac{s_{t_2}^1 - s_{t_2}^0}{s_{t_1}^1 - s_{t_1}^0} \right) \|$ is the 2-norm of the matrix of which the element $ij$ is: $\mathbb{E} \left( \frac{s_{t_2}^1(i) - s_{t_2}^0(i)}{s_{t_1}^1(j) - s_{t_1}^0(j)} \right)$, where $s_t(i)$ is the i-th element in the vector $s_t$.*

**Assumption 3** (Bounded region). *There exist a finite $D < \infty$ such that: (Wu et al., 2020; Wang et al., 2017)*

$$\| s_1 - s_2 \| \leq D \text{ for } \forall s_1, s_2 \in \mathcal{S}. \tag{14}$$

**Assumption 4** (Lipschitz). *The transition model and the reward model are both Lipschitz continuous with Lipschitz constant $L_f$ and $L_r$ respectively (Clavera et al., 2019). In addition, we need the derivative of the transition model and the reward model to be Lipschitz continuous with Lipschitz constant $L_{fg}$ and $L_{rg}$.*

**Assumption 5** (Uniform ergodicity). *For a fixed policy $\pi_\theta(\cdot|s_0)$, denote $\mu_\theta$ as the stationary distribution induced by the policy and the transition probability measure $P(\cdot|s, a)$. Then there exists $m < \infty$, $C_0 < \infty$, and $\rho \in (0,1)$ such that:(Wu et al., 2020; Zhang et al., 2020a; Qiu et al., 2021)*

$$d_{TV} \left( P(s_{t+\tau} \in \cdot | s_t = s), \mu_\theta(\cdot) \right) \leq m\rho^\tau \; \forall \tau \geq 0, \; \forall s \in \mathcal{S}. \tag{15}$$

**Assumption 6** (Lipschitz policy). *Let $\pi_\phi(a|s)$ be a policy parameterized by $\theta$. there exist constants such that for all given state $s$ and action $a$ we have:(Wu et al., 2020; Clavera et al., 2019; D' Oro & Jaśkowski, 2020)*

$$\left\| \frac{d\pi_\phi(a|s)}{d\phi} \right\| \leq L_\pi. \tag{16}$$

**Assumption 7** (Step-size). *The step-size $\alpha_t$ is non-increasing and non-negative (Wang et al., 2017; Wu et al., 2020).*

**Assumption 8** (Bounded reward). *The reward is bounded in the region $[0, r_{max}]$ (Clavera et al., 2019; Wu et al., 2020).*

Here, we show some lemmas that is important for the proof.

**Lemma 1.** *Suppose assumptions hold, denote the $C_0 = \min(\frac{D}{\epsilon_f}, \frac{1 - L_f^{t+1}}{1 - L_f})$ for $\forall t$ we have*

$$\| \tilde{s}_t - s_t \| \leq C_0 \epsilon_f \tag{17}$$

**Lemma 2.** *Denote $C_1 = tL_1^2$, $C_2 = tC_0 L_{fg} L_1^2$. We have*

$$\delta \left( \mathbb{E} \left( \frac{ds_t}{ds_0} \right) \right) = \left\| \mathbb{E} \left( \widehat{\frac{ds_t}{ds_0}} - \frac{ds_t}{ds_0} \right) \right\| \leq C_1 \epsilon_f^g + C_2 \epsilon_f \tag{18}$$

**Lemma 3.** *If consent $0 < \gamma < 1$, for arbitrary constant $C$, we have:*

$$\sum_{t=0}^{\infty} t\gamma^t C = \frac{\gamma}{(1-\gamma)^2} C \tag{19}$$

**Lemma 4.** *Denote constants $C_3 = \frac{L_1}{1-\gamma}$ and $C_4 = \frac{L_r L_1^2 \gamma}{(1-\gamma)^2}, C_5 = \frac{C_0 L_1 L_{rg}}{(1-\gamma)} + \frac{C_0 L_{fg} L_r \gamma}{(1-\gamma)^2}$, then we have:*

$$\delta \left( \frac{dR}{ds} \right) = \left\| \mathbb{E} \left( \frac{dR^{\widehat{M}}}{ds} - \frac{dR^M}{ds} \right) \right\| \leq C_3 \epsilon_r^g + C_4 \epsilon_f^g + C_5 \epsilon_f \tag{20}$$

**Lemma 5.** *Considering a Markov process $M$ with stationary distribution $\mu$. Data $\xi_t$ are sampled from the $M$. There are two functions of parameter $\phi$: $G(\phi_t, \xi_t)$ and $g(\phi_t)$ where*

$$g(\phi_t) = \mathbb{E}_{\xi \sim \mu} G(\phi_t, \xi). \tag{21}$$

*Denote the upper bound for the function $G$ as $L_1$. Denote the Lipschitz constant for the function $G$ as $L_2$.*

$$\phi_{t+1} = \phi_t - \alpha_t G(\phi_t, \xi_t) \tag{22}$$

*For all $0 < T < \infty$, we can proof that:*

$$\left\| \sum_{t=1}^{T} \alpha_t \langle g(\phi_t), G(\phi_t, \xi_t) - g(\phi_t) \rangle \right\| \leq 2\tau \alpha_0 L_1^2 + \sum_{t=\tau+1}^{T} \alpha_t^2 L_1^2 L_2 \tau + \sum_{t=1}^{T} \alpha_t L_1 m \rho^\tau \tag{23}$$

**Lemma 6.** *Denote $U_J$ as the upper bound of the expected sum of discounted rewards $J$. Then $U_J \leq \frac{r_{max}}{1-\gamma}$*

**Lemma 7.** *There exist two constants $L_J$ and $L_J^g$ such that for all $\phi_1, \phi_2$, we have*

$$\|J(\phi_1) - J(\phi_2)\| \leq L_J \|\phi_1 - \phi_2\| \tag{24}$$

$$\left\| \frac{dJ}{d\phi_1} - \frac{dJ}{d\phi_2} \right\| \leq L_J^g \|\phi_1 - \phi_2\| \tag{25}$$

**Lemma 8.** *There exist constants $L_R$ such that for all $\phi_1, \phi_2$, we have*

$$\mathbb{E} \|R(\phi_1) - R(\phi_2)\| \leq L_R \|\phi_1 - \phi_2\| \tag{26}$$

$$\tag{27}$$

**Lemma 9.** *Suppose the function $J$ is smooth with smoothness constant $L_J^g$*

$$J(\phi_2) \geq J(\phi_1) + \langle \frac{dJ}{d\phi_1}, \phi_2 - \phi_1 \rangle - \frac{L_J^g}{2} \|\phi_2 - \phi_1\| \tag{28}$$

**Lemma 10.** *If $x, y, z > 0$ and $\sqrt{x} \leq \sqrt{y} + \sqrt{z}$, then we have*

$$x \leq 2y + 2z \tag{29}$$

### C.1 PROOF OF THEOREM 1

Overall speaking, the main idea behind the proof of Theorem 1 can be summarized as follows.

Considering that to calculate a gradient value of a function output w.r.t its input, we need to specify the function and the input value. Therefore, as in the Equation 3, both the function($\tilde{f}$ and $\tilde{r}$) to be taking gradient and the input value $\tilde{s}_t$ are different from the ground-truth. So we need to first bound the error of the prediction state(Lemma 1) and then bound the error of the gradient of the learned model in true state (Lemma 4).

Firstly, we define some notations, denote $C_6 = \frac{L_f L\pi + L_f C_3 L\pi\gamma}{1-\gamma}$, $C_7 = \frac{\gamma L_f L_\pi C_4 + L_1 L_\pi \gamma/(1-\gamma)}{1-\gamma}$, $C_8 = \frac{C_0 L_{r^g} + \gamma C_5 L_f L_\pi + L_1 L_r L_\pi L_{f^g} C_0 \gamma/(1-\gamma)}{1-\gamma}$, now we can prove the theorem by using previous lemmas:

$$\left\| \mathbb{E} \left( \frac{dR^{\widehat{M}}}{d\phi} - \frac{dR^M}{d\phi} \right) \right\| \tag{30}$$

$$= \left\| \mathbb{E} \left( \sum_{t=0}^{\infty} \gamma^t \left( \frac{\widehat{d\tilde{r}_t}}{d\phi} - \frac{dr_t}{d\phi} \right) + \gamma \sum_{t=0}^{\infty} \gamma^t \left( \frac{\widehat{dR}}{d\tilde{s}_{t+1}} \frac{\widehat{d\tilde{s}_{t+1}}}{d\tilde{a}_t} \frac{d\tilde{a}_t}{d\phi} - \frac{dR}{ds_{t+1}} \frac{ds_{t+1}}{da_t} \frac{da_t}{d\phi} \right) \right) \right\| \tag{31}$$

$$\leq \frac{1}{1-\gamma} (L_{r^g} C_0 \epsilon_f + \epsilon_r^g) L_\pi + \left\| \mathbb{E} \left( \gamma \sum_{t=0}^{\infty} \gamma^t \left( \delta \left( \frac{dR}{ds} \right) \frac{\widehat{d\tilde{s}_{t+1}}}{d\tilde{a}_t} + \frac{dR}{ds_{t+1}} \delta \left( \frac{ds}{da} \right) \right) L_\pi \right) \right\| \tag{32}$$

$$\leq \frac{1}{1-\gamma} (L_{r^g} C_0 \epsilon_f + \epsilon_r^g) L_\pi + \frac{\gamma}{1-\gamma} L_f L_\pi (C_3 \epsilon_r^g + C_4 \epsilon_f^g + C_5 \epsilon_f) + \frac{\gamma}{(1-\gamma)^2} L_1 L_r L_\pi (L_{f^g} C_0 \epsilon_f + \epsilon_f^g) \tag{33}$$

$$= C_6 \epsilon_r^g + C_7 \epsilon_f^g + C_8 \epsilon_f. \tag{34}$$

The first equation is the definition. The second inequality follows the assumption 2, the right term . The third term follows the lemma 4

**Remark:** This theorem is fit for the case that the Lipschitz constant $L_f < 1$. For the dynamics that the gradient is always larger than 1, it is unrealistic and uncontrollable. For the dynamics that the gradient is always less than 1, theorem 1 and the assumptions are enough. For the last case, it needs a stronger assumption directly on $\|\tilde{s}_t - s_t\|$.

## C.2 PROOF OF THE THEOREM 2

First of all, we briefly summarize the proof idea here. We first decompose the difference of objective function brought by parameter updating into two terms according to its smooth assumption(Equation 35). The first term is an inner product term. The inter product term can be further decompose into two terms(Equation 36). One is about the model error and the other one is about the non-i.i.d. data. We can carefully bound each terms and then combine them all together to proof the theorem. Readers can refer to the related papers Wang et al. (2017) Wu et al. (2020) Ghadimi & Lan (2013) to get more insight of the proof.

According to the lemma 9, we can have:

$$J(\phi_{t+1}) \geq J(\phi_t) + \alpha_t \left\langle \frac{dJ}{d\phi_t}, \frac{dR^{\hat{M}}}{d\phi} \right\rangle - \frac{L_J}{2} \|\phi_{t+1} - \phi_t\|^2, \tag{35}$$

where $\frac{dR^{\hat{M}}}{d\phi}$ is the estimated policy gradient calculated using learned model and used to update the policy parameter from $\phi_t$ to $\phi_{t+1}$.

Denote $h_t^1 = \frac{dR^{\hat{M}}}{d\phi} - \frac{dR^M}{d\phi}, h_t^2 = \frac{dR^M}{d\phi}$.

We can further decompose the value improvement as

$$J(\phi_{t+1}) \geq J(\phi_t) + \alpha_t \left( \left\langle \frac{dJ}{d\phi_t}, h_t^1 \right\rangle + \left\langle \frac{dJ}{d\phi_t}, h_t^2 \right\rangle \right) - \frac{L_J^g}{2} \|\phi_{t+1} - \phi_t\|^2. \tag{36}$$

According to the Cauchy–Schwarz inequality we have

$$\mathbb{E} \left\langle \frac{dJ}{d\phi_t}, h_t^1 \right\rangle \geq -\sqrt{E \left\| \frac{dJ}{d\phi_t} \right\|^2 \mathbb{E} \|h_t^1\|^2} \geq -\sqrt{E \left\| \frac{dJ}{d\phi_t} \right\|^2} (C_6 \epsilon_r^g + C_7 \epsilon_f^g + C_8 \epsilon_f). \tag{37}$$

Rearrange the term we have

$$\mathbb{E} \left\langle \frac{dJ}{d\phi_t}, h_t^2 \right\rangle = \mathbb{E} \left\langle \frac{dJ}{d\phi_t}, \frac{dR^M}{d\phi_t} - \frac{dJ}{d\phi_t} \right\rangle + \mathbb{E} \left\langle \frac{dJ}{d\phi_t}, \frac{dJ}{d\phi_t} \right\rangle. \tag{38}$$

Note that the first term in the right hand side is 0 if the data is I.I.D. sampled from the stationary distribution. If the data is sampled from the markov process, the data is non-i.i.d.. we can further bound the first term using the Lemma 5

$$\sum_{t=1}^{T} \alpha_t \mathbb{E} \left\langle \frac{dJ}{d\phi_t}, \frac{dR^M}{d\phi_t} - \frac{dJ}{d\phi_t} \right\rangle \geq - \left( 2\tau L_J \alpha_0 + \sum_{t=1}^{T} \alpha_t^2 L_J^2 L_J^g \tau + \sum_{t=1}^{T} \alpha_t L_J m \rho^\tau \right). \tag{39}$$

The second term is

$$\mathbb{E} \left\langle \frac{dJ}{d\phi_t}, \frac{dJ}{d\phi_t} \right\rangle = \mathbb{E} \left\| \frac{dJ}{d\phi_t} \right\|^2. \tag{40}$$

Noticing that

$$\phi_{t+1} - \phi_t = \alpha_t \frac{dR^{\hat{M}}}{d\phi_t}. \tag{41}$$

Combine Equations (37), (39), (40), (41), we can get

$$\alpha_t B_1 - \alpha_t B_3 \sqrt{B_1} \le J(\phi_{t+1}) - J(\phi_t) + \alpha_t B_4 + B_5, \tag{42}$$

where $B_1 = \left\| \frac{dJ}{d\phi_t} \right\|^2$, $B_2 = 2\tau L_J \alpha_0 + \sum_{t=1}^{T} \alpha_t^2 \left( L_J^2 L_J^g \tau + \frac{L_J^g L_R^2}{2} \right) + \sum_{t=1}^{T} \alpha_t L_J m \rho^\tau$, $B_3 = C_6 \epsilon_r^g + C_7 \epsilon_f^g + C_8 \epsilon_f$, $B_4 = \mathbb{E} \left\langle \frac{dJ}{d\phi_t}, \frac{dR^M}{d\phi_t} - \frac{dJ}{d\phi_t} \right\rangle$ $B_5 = \frac{L_J^g \alpha_t^2 L_R^2}{2}$ Notice $B_1, B_4, B_5$ are functions of $t$.

Taking summarizing on both side, we have

$$\sum_{t=1}^{T} \alpha_t B_1 - \sum_{t=1}^{T} \alpha_t B_3 \sqrt{B_1} \le J(\phi_{T+1}) - J(\phi_0) + \sum_{t=1}^{T} \alpha_t B_4 + \sum_{t=1}^{T} B_5, \tag{43}$$

$$\frac{\sum_{t=1}^{T} \alpha_t B_1}{\sum_{t=1}^{T} \alpha_t} - \frac{\sum_{t=1}^{T} \alpha_t B_3 \sqrt{B_1}}{\sum_{t=1}^{T} \alpha_t} \le \frac{2U_J + B_2}{\sum_{t=1}^{T} \alpha_t}. \tag{44}$$

We have

$$\frac{\sum_{t=1}^{T} \alpha_t (\sqrt{B_1} - \frac{B_3}{2})^2}{\sum_{t=1}^{T} \alpha_t} \le \frac{2U_J + B_2}{\sum_{t=1}^{T} \alpha_t} + \frac{B_3^2}{4}. \tag{45}$$

Therefore,

$$\min_{1 \le k \le T} (\sqrt{B_1} - \frac{B_3}{2})^2 \le \frac{2U_J + B_2}{\sum_{t=1}^{T} \alpha_t} + \frac{B_3^2}{4}. \tag{46}$$

According to the lemma 10 We can prove that

$$\min_{1 \le k \le T} B_1 \le \frac{4U_J + 2B_2}{\sum_{t=1}^{T} \alpha_t} + B_3^2. \tag{47}$$

### C.3 PROOF OF THE USEFUL LEMMAS

*Proof of Lemma 1.* According to the transition model definition and its Lipschitz continuous assumption, on the one hand, since the states are bounded in the range with diameter $D$, we can see that $\|\tilde{s}_t - s_t\| \le D$. On the other hand,

$$\|\tilde{s}_t - s_t\| \le \epsilon_f + L_f \|\tilde{s}_{t-1} - s_{t-1}\| \le \left( \frac{1 - L_f^{t+1}}{1 - L_f} \right) \epsilon_f. \tag{48}$$

$\square$

*Proof of the Lemma2.*

$$\left\| \mathbb{E} \left( \frac{\widehat{d\tilde{s}_t}}{d\tilde{s}_{t-1}} \cdot \ldots \cdot \frac{\widehat{d\tilde{s}_1}}{d\tilde{s}_0} \right) - \mathbb{E} \left( \frac{ds_t}{ds_{t-1}} \cdot \ldots \cdot \frac{ds_1}{ds_0} \right) \right\| \tag{49}$$

$$= \left\| \mathbb{E} \sum_{i=1}^{t} \left( \Pi_{j=0}^{i-1} \frac{ds_j}{ds_{j-1}} \right) \left( \frac{\widehat{d\tilde{s}_i}}{d\tilde{s}_{i-1}} - \frac{ds_i}{ds_{i-1}} \right) \left( \Pi_{j=i+1}^{t} \frac{\widehat{d\tilde{s}_j}}{d\tilde{s}_{j-1}} \right) \right\| \tag{50}$$

$$\le \mathbb{E} \sum_{i=1}^{t} \left( \left\| \Pi_{j=0}^{i-1} \frac{ds_j}{ds_{j-1}} \right\| \right) \left( \left\| \frac{\widehat{d\tilde{s}_i}}{d\tilde{s}_{i-1}} - \frac{\widehat{ds_i}}{ds_{i-1}} \right\| + \left\| \frac{\widehat{ds_i}}{ds_{i-1}} - \frac{ds_i}{ds_{i-1}} \right\| \right) \left( \left\| \Pi_{j=i+1}^{t} \frac{\widehat{d\tilde{s}_j}}{d\tilde{s}_{j-1}} \right\| \right) \tag{51}$$

$$\le \sum_{i=1}^{t} L_1^2 (\epsilon_f^g + C_0 \epsilon_f L_{f^g}) \tag{52}$$

$$\le t L_1^2 \epsilon_f^g + t C_0 L_{f^g} L_1^2 \epsilon_f. \tag{53}$$

$\square$

*Proof of the Lemma 3.* Let $s = \sum_{t=0}^{\infty} t\gamma^t C$, then we can get that $\gamma s = \sum_{t=0}^{\infty} t\gamma^{t+1}C = \sum_{t=1}^{\infty}(t-1)\gamma^t C$. So, $s - \gamma s = (1-\gamma)s = \sum_{t=1}^{\infty} \gamma^t C = \frac{\gamma}{1-\gamma}C$. At last we can get $s = \frac{\gamma}{(1-\gamma)^2}C$. $\qquad\square$

*Proof of the Lemma 4.*

$$\left\| \mathbb{E}\left( \frac{dR^{\widehat{M}}}{ds} - \frac{dR^M}{ds} \right) \right\| = \left\| \mathbb{E}\sum_{t=0}^{\infty} \gamma^t \frac{\widehat{d\tilde{r}_t}}{d\tilde{s}_t}\frac{\widehat{d\tilde{s}_t}}{\tilde{s}_0} - \mathbb{E}\sum_{t=0}^{\infty}\gamma^t \frac{dr_t}{ds_t}\frac{ds_t}{s_0} \right\| \tag{54}$$

$$\leq \mathbb{E}\sum_{t=0}^{\infty}\gamma^t \left( \left\| \frac{\widehat{d\tilde{s}_t}}{d\tilde{s}_0}\delta\left( \frac{dr_t}{ds_t} \right) \right\| + \left\| \frac{dr_t}{ds_t}\delta\left( \frac{ds_t}{ds_0} \right) \right\| \right) \tag{55}$$

$$\leq \sum_{t=0}^{\infty}\gamma^t \left( L_1(\epsilon_r^g + L_{r^g}C_0\epsilon_f) + L_r\left( C_1\epsilon_f^g + C_2\epsilon_f \right) \right) \tag{56}$$

$$= \frac{1}{1-\gamma}\left( L_1(\epsilon_r^g + L_{r^g}C_0\epsilon_f) \right) + L_r\sum_{t=0}^{\infty}\gamma^t \left( C_1\epsilon_f^g + C_2\epsilon_f \right) \tag{57}$$

$$= \frac{1}{1-\gamma}\left( L_1(\epsilon_r^g + L_{r^g}C_0\epsilon_f) \right) + \frac{\gamma}{(1-\gamma)^2}L_r\epsilon_f^g L_1^2 + \frac{\gamma}{(1-\gamma)^2}C_0 L_{f^g}L_r\epsilon_f \tag{58}$$

$$= C_3\epsilon_r^g + C_4\epsilon_f^g + C_5\epsilon_f. \tag{59}$$

$\qquad\square$

*Proof of the Lemma 5.*

$$\sum_{t=1}^{T}\alpha_t \langle g(\phi_t), G(\phi_t, \xi_t) - g(\phi_t)\rangle \tag{60}$$

$$= \sum_{t=1}^{T}\alpha_t \left( \langle g(\phi_t), G(\phi_t, \xi_t) - G(\phi_t, \xi_{t+\tau})\rangle + \langle g(\phi_t), G(\phi_t, \xi_{t+\tau}) - g(\phi_t)\rangle \right). \tag{61}$$

We first bound the first term in the right hand side.

$$\| \sum_{t=1}^{T}\alpha_t \left( \langle g(\phi_t), G(\phi_t, \xi_t) - G(\phi_t, \xi_{t+\tau})\rangle \right)\| \tag{62}$$

$$\leq \| \sum_{t=1}^{\tau}\alpha_t \langle g(\phi_t), G(\phi_t, \xi_t)\rangle\| + \| \sum_{t=T-\tau+1}^{T}\alpha_t \langle g(\phi_t), G(\phi_t, \xi_{t+\tau})\rangle\| \tag{63}$$

$$+ \| \sum_{t=\tau+1}^{T}\alpha_t \langle g(\phi_t), G(\phi_t, \xi_t) - G(\phi_{t+\tau}, \xi_t)\rangle\| \tag{64}$$

$$\leq 2\tau\alpha_0 L_1^2 + \| \sum_{t=\tau+1}^{T}\alpha_t \langle g(\phi_t), G(\phi_t, \xi_t) - G(\phi_{t+\tau}, \xi_t)\rangle\| \tag{65}$$

$$\leq 2\tau\alpha_0 L_1^2 + \sum_{t=\tau+1}^{T}\alpha_t \| \sum_{k=t}^{t+\tau-1} \langle g(\phi_t), G(\phi_k, \xi_t) - G(\phi_{k+1}, \xi_t)\rangle\| \tag{66}$$

$$\leq 2\tau\alpha_0 L_1^2 + \sum_{t=\tau+1}^{T}\alpha_t L_2 \| \sum_{k=t}^{t+\tau-1} \langle g(\phi_t), \phi_k - \phi_{k+1}\rangle\| \tag{67}$$

$$\leq 2\tau\alpha_0 L_1^2 + \sum_{t=\tau+1}^{T}\alpha_t^2 L_1^2 L_2\tau. \tag{68}$$

$$\tag{69}$$

Now, we will bound the second term in equation in expectation.

$$\left\| \mathbb{E} \left( \sum_{t=1}^{T} \alpha_t \left( \langle g(\phi_t), G(\phi_t, \xi_{t+\tau}) - g(\phi_t) \rangle \right) \right) \right\| \tag{70}$$

$$= \sum_{t=1}^{T} \alpha_t \left( \langle g(\phi_t), \mathbb{E} \left( G(\phi_t, \xi_{t+\tau}) - g(\phi_t) | \mathcal{F}_t \right) \rangle \right) \tag{71}$$

$$\leq \sum_{t=1}^{T} \alpha_t L_1 \left\| \int G(\phi_t, \xi) \left( P(\xi_{t+\tau} \in \xi | \mathcal{F}_t) - \mu(\xi) \right) d\xi \right\| \tag{72}$$

$$\leq \sum_{t=1}^{T} \alpha_t L_1 d_{TV} \left( P(\xi_{t+\tau} \in \cdot | \mathcal{F}_t), \mu(\cdot) \right) \tag{73}$$

$$\leq \sum_{t=1}^{T} \alpha_t L_1 m \rho^{\tau}. \tag{74}$$

$\square$

*Proof of the Lemma 6.*

$$J = \mathbb{E} \sum_{t=1}^{\infty} \gamma^t r_t \leq \frac{r_{max}}{1 - \gamma} = U_J. \tag{75}$$

$\square$

*Proof of the Lemma 7.* The first inequality comes from Lemma B.4 in Wu et al. (2020). The second inequality comes from Lemma 3.2 in Zhang et al. (2020a).

$\square$

*Proof of the Lemma 8.*

$$\left\| \mathbb{E} \left( \frac{dR^M}{ds} \right) \right\| = \left\| \mathbb{E} \sum_{t=0}^{\infty} \gamma^t \frac{dr_t}{ds_t} \frac{ds_t}{s_0} \right\| \leq \mathbb{E} \sum_{t=0}^{\infty} \gamma^t \left\| \frac{dr_t}{ds_t} \frac{ds_t}{s_0} \right\| \leq \sum_{t=0}^{\infty} \gamma^t L_1 L_r = \frac{1}{1 - \gamma} L_1 L_r. \tag{76}$$

$$\left\| \mathbb{E} \frac{dR^M}{d\phi} \right\| = \left\| \mathbb{E} \left( \sum_{t=0}^{\infty} \gamma^t \left( \frac{dr_t}{d\phi} \right) + \gamma \sum_{t=0}^{\infty} \gamma^t \left( \frac{dR}{ds_{t+1}} \frac{ds_{t+1}}{da_t} \frac{da_t}{d\phi} \right) \right) \right\| \tag{77}$$

$$\leq \frac{1}{1 - \gamma} L_r L_\pi + \left\| \mathbb{E} \gamma \sum_{t=0}^{\infty} \gamma^t \left( \frac{dR}{ds_{t+1}} \frac{ds_{t+1}}{da_t} \frac{da_t}{d\phi} \right) \right\| \leq \frac{1}{1 - \gamma} L_r L_\pi + \frac{\gamma}{(1 - \gamma)^2} L_1 L_\pi L_f L_r \tag{78}$$

$$= L_R. \tag{79}$$

$\square$

*Proof of the Lemma 9.* The conclusion comes from the Equation 4.4.9 in Nesterov et al. (2018). $\square$

*Proof of the Lemma 10.* Since $\sqrt{x}$ and $\sqrt{y} + \sqrt{z}$ are both positive, then we have $x \leq (\sqrt{y} + \sqrt{z})^2$, i.e., $x \leq y + z + 2\sqrt{yz}$, note that $2\sqrt{yz} \leq y + z$, then we have $x \leq 2y + 2z$. $\square$

