# OpenReview forum: "Gradient Information Matters in Policy Optimization by Back-propagating through Model"
_ICLR.cc/2022/Conference — ICLR 2022 Poster_

### Official Review · Reviewer_9ZXt · 2021-11-01

**Correctness:** 4
**Technical Novelty And Significance:** 3
**Empirical Novelty And Significance:** 3
**Recommendation:** 8
**Confidence:** 4

**Main Review:**

**Originality:** The method presented is quite original and well-positioned among existing methods for model learning.

**Significance:** The problem in object is of interest for the RL community.

**Rigour:** The theoretical results are not of particular benefit for the practical algorithm. The theoretical results and experimental protocols seem sound, although the core change proposed by the method does not provide very large improvements over baselines.

**Strengths**

- Directly learning the Jacobians of the dynamics is a very appealing goal, given the resurgence in popularity of value gradient methods (i.e., methods which compute the policy gradient by direct differentiation through a learned model).
- The presented approach is well-positioned in the literature as a *decision-aware* approach for model learning.
- The theoretical results presented in Section 4, despite not being novel in essence, are quite interesting.
- Both the proposed loss function for model learning and the idea of having two models for trajectory generation and gradient computation are, as far as I know, new and simple to implement.

**Major Concerns**

- The loss function in Equation 12 requires to select the nearest neighbours of the current data point. With the usual implementation for the replay buffer, this adds a significant computational burden to the already expensive model-based policy optimization algorithm. Is the method directly facing this issue? What is the actual computational complexity of the algorithm?
- It seems, especially from Figure 4 in Appendix, that a large part of the additional performance of the proposed method comes from the two-models component and not from the gradient learning one. Moreover, there is the risk that much of the improvement provided by the use of two models actually comes from the additional capacity. A good experiment should control for the number of parameters to check whether the performance boost only comes from the additional parameters, and not the particular architecture in use. Thus, I would have expected to see an experiment in which the number of parameters used by the two models in total is the same as the one used by the baseline with a model alone.
- To double check more robustly whether the proposed objective for gradient estimation is effective per se, it would be extremely beneficial to have a simple, independent, experiment, in a supervised learning problem.
- The mechanism used for backpropagating through a model after generating a trajectory from another one is unclear. To the best of my knowledge, the only technique for achieving something similar while using stochastic models is the one used in SVG (Heess et al., 2015) to differentiate through trajectories of real experience via reparameterization. This should be clearly explained in the paper.
- In Figure 3, a more appropriate baseline to show should be MAAC, which seems to be the core algorithm upon which the approach is based, given its backpropagation through the learned model.

**Minor Concerns**

- I believe it would be more clear to state the assumptions before stating the two theoretical results, e.g., in the background section.
- It could be beneficial to explain the loss function in Equation 12 in terms of a Taylor approximation: the learned model should have a gradient such that its first-order approximation is consistent with the given data. Is this a valid interpretation?
- Section 4 should contain an explanation of why the gradient loss in Equation 12 alone would be insufficient for training a model. Are both of them used to avoid degenerate solutions only, or is there a deeper motivation?

**Summary Of The Paper:**

Many recent model-based reinforcement learning algorithms exploit the differentiability of the learned model by computing gradients through it. Nonetheless, the standard approach for model learning is to minimize a loss on the prediction of the next environment state which, when the underlying dynamics is complex, does not guarantee that also the resulting gradient information will be accurate. The paper proposes a method for model learning which explicitly tries to estimate the Jacobians of the underlying dynamics, by leveraging a loss function on the directional derivative and a nearest neighbour approximation. Additionally, the paper proposes to use a model learned with a combination of this new loss and a prediction loss for backpropagation only, while training a traditional model for the generation of trajectories.

**Summary Of The Review:**

There are some issues in the experimental results and in the justification of some moving parts of the algorithm, but the paper proposes a simple and seemingly sound solution for a very important problem in model-based reinforcement learning. I thus lean towards recommending to accept it.

---
The authors addressed my main concerns. Thus, I now recommend acceptance.

---

> ### Author Response · Authors · 2021-11-18
> **To Reviewer 9ZXt**
>
> Thank you for your positive comments on our work. We revise our paper according to your suggestions and provide our response to your concern below.
>
> **1.	Your concern about the efficiency of computing the n-nearest points in the replay buffer**
> -	In practice, we use KD-tree to construct and search the top-n nearest points in the buffer for each point. The averaged search and insert complexity are both $O (log_2 N) $ .
> -	The model learning procedure is the same as MBPO [1] that trains the model every 250 env-steps and uses early stop. We just need to search $n$ nearest points for each point in the buffer before the model training begins. Specifically, we first sample $min(100000, buffer\ size)$ points, and then for each point in the buffer,  we search the $n$ nearest points in this sampled dataset. So, the complexity will not go explosion.
> -	The running time for building KD-tree and searching n nearest points one time is 7.7s for 50k samples and 8.32s for 100k samples in hopper(dim=14). Please note that we only need to do building and searching 4 times in 1k environment step.
> **2.	Do these extra parameters account for the performance difference**
> -	The answer is NO.
> -	To address the concern, we additionally add experiments in our revised version that the number of parameters used by the two models in total is the same as the one used by the baseline with a model alone. The lines labeled `baseline(#param ~2M)` in ablation mean learning the model without gradient information but using a bigger number of parameters. From the results, we can see that directly added parameters don't make much improvement which yields the value of our method.  Besides these, we will add experiments that all baselines using a model with a bigger size when finished.
> **3.	Independent supervised learning experiments**
> -	We think section 5.2 already addresses your concern. The curves in Figure 2 show how gradient error changes during training. We can know that model trained with the gradient error loss can learn a more precise gradient estimator.
> **4.	 The mechanism used for backpropagating through a model**
> -	Our model is used the same as MBPO, which acts as a deterministic predictor (use the output, mean, as the prediction), so we don't need to use the tricks from SVG. To make this part clearer, we have additionally added a part in the appendix B.1 to give more information in our revised version.
> **5.	MAAC baseline**
> -	We adjust figure 3 that use the results of MAAC as a baseline.
> **6.	Assumption**
> -	We summarize the necessary assumption and put it before the theorem. We put all the assumptions into the appendix before the proof of the main theorem.
> **7.	Explanation for Equation 12**
> -	Interpreting it as a first-order Taylor approximation is correct and easy to understand. We are more than happy to take your advice to revise our paper.
> **8.	Why the gradient loss in Equation 12 alone would be insufficient**
> -	Theoretically, estimating a gradient vector needs at least n orthogonality directional derivatives. However, we can just use the data in the buffer to estimate the directional derivatives. So, if we only sample m $(<n)$ samples to calculate the loss in Equation 12, there will be more than one local optimal.  To tradeoff the accuracy and the computational complexity, we proposed to incorporate the prediction loss to constrain the hypothesis space of the learned model to prevent large bias local optimal.

---

> > ### Comment · Reviewer_9ZXt · 2021-11-21
> > **Thanks for your answer — I still don’t get how the two models are used for backpropagation**
> >
> > Thank you so much for addressing most of my and the other reviewers’ concerns. I think this helped in improving the paper.
> >
> > I am still not sure of how the gradient and prediction model are used exactly, even after reading the description you have added in Appendix. If the two models are used deterministically, but trained differently, the next states produced by the two models when given a tuple $(s,a)$ as an input will be different, so the two computational graphs will be different and you can’t use the second model for backpropagating through the graph produced by the first one.
> >
> > Do you mean that the gradient is summed from both the models when computing the policy gradient but the critic is only learned by using transitions generated by the prediction model (this does not seem to be the case from the pseudo code)? Or is there something that I miss as a method for backpropagating with a deterministic model through the trajectories generated by another deterministic model?

---

> > > ### Author Response · Authors · 2021-11-23
> > > **Thanks for your feedback! How the two models are used for backpropagation**
> > >
> > > Thank you for continuing the discussion and your opinion helps improve the paper.
> > >
> > > - Firstly, we don't mean that the gradient is summed from both models. Please note that, for a function $f$, to calculate the gradient value $df(x)/dx$, we just need to know the input value $x$ and the function formulation $f$. In our case, we use the current policy to generate the trajectory by interaction with the true model and the learned prediction model and use the learned gradient model to calculate $ds'/ds$ when needed.
> > >
> > >     - Let's look at a toy example. The true model is $M$. The gradient model is $M^g$. The prediction model is $M^p$. The current policy is $\pi_\phi$. We first interacting with true model $M$ to get the $\{s_0,a_0,r_0\}$ and then roll out $\pi_{\phi}$ on $M^p$  to get $\{s_0,a_0,r_0,s_1,a_1,r_1,s_2\}$. Now we need to calculate $\frac{d(r_0+r_1)}{d\phi}$. It can be written as the following form by chain rule:
> > > $\frac{d(r_0+r_1)}{d\phi}=\frac{dr_0}{da_0}\frac{da_0}{d\phi}+\frac{dr_1}{da_1}\frac{da_1}{d\phi}+(\frac{dr_1}{ds_1}+\frac{dr_1}{da_1}\frac{da_1}{ds_1})\frac{ds_1}{da_0}\frac{da_0}{d\phi}.$
> > >
> > >     - For the factors that do not related to the gradient of model($\frac{da_1}{ds_1}, \frac{da_0}{d\phi}, \frac{da_1}{d\phi}$),  we can directly calculate them.
> > >  For other factors that related to the gradient of the model, such as $\frac{dr_0}{da_0}$ and $\frac{ds_1}{da_0}$, we use $\frac{dM^g(s_0,a_0)}{da_0}$ as an estimator for ground truth $\frac{dM(s_0,a_0)}{da_0}$. And $\frac{dM^g(s_0,a_0)}{da_0}$ is calculated by taking ${s_0,a_0}$ as input to the gradeint model and back-propagating. Combined with the saved part we get the final gradient we want.
> > > We use $\frac{dM^g(s_0,a_0)}{da_0}$ rather than $\frac{dM^p(s_0,a_0)}{da_0}$ since we believe $\frac{dM^g(s_0,a_0)}{da_0}$ is closer to the ground truth $\frac{dM(s_0,a_0)}{da_0}$.
> > >
> > > - Secondly, the critic is only learned by using transitions generated by the prediction model (corresponding to lines 7-15 in pseudo-code) since the prediction model can generate transitions more closely to the ground truth and we don't need the gradient here. This is also supported by the results in Figure 6 that the predictive error of the prediction model $M^p$ is smaller than the gradient model $M^g$. This is also an intuitive explanation of why we use two models respectively.
> > >
> > > At last, we provide an implementation in [github]( https://github.com/paperddppo/ddppo/tree/main/ddppo), hope this can help understand the procedure of our work, thank you.

---

> > > > ### Comment · Reviewer_9ZXt · 2021-11-23
> > > > **States are different, but if the approach works in practice...**
> > > >
> > > > Thank you for your explanation.
> > > >
> > > > The output state of the two models will still be different, so the Jacobian evaluated with the gradient model will be considering a different state transition. This is a source of additional bias, especially for long trajectories; but, given the empirical nature of this proposed technique, and the fact that the model is already introducing its own bias into the estimation of the policy gradient, I am okay with it causing the improvement in performance.

---

> > > > > ### Author Response · Authors · 2021-11-24
> > > > > **Thanks for your positive feedback on our work.**
> > > > >
> > > > > - Thanks for your positive feedback on our work. Your comments help us much when preparing the revised version of the manuscript.
> > > > > - We keep polishing our work to make the result more convictive and the theory more solid.
> > > > > - Please feel free to discuss if there is anything else to clarify.

---

### Official Review · Reviewer_8zAV · 2021-11-01

**Correctness:** 3
**Technical Novelty And Significance:** 3
**Empirical Novelty And Significance:** 2
**Recommendation:** 6
**Confidence:** 4

**Main Review:**

## UPDATE DURING REBUTTAL

To AC & reviewers & authors - I think the authors did make substantial improvements to the paper and it is now easier to understand. Given the prevalence of this problem in MBRL I do think this is a contribution to help the field understand how best to learn dynamics models for control. Though, the empirical questions make my recommendation very borderline. My idea is that it is closer to a weak accept now, but it will be hard to fight for this paper's acceptance without a little more understanding of why it works.

## General Comments
- **technical merit**: the core idea the authors to improve model-based reinforcement learning are well motivated. The proposed solution is interesting and novel. Though, they seem to not be citing a core paper motivating their work: Lambert, Nathan, et al. "Objective Mismatch in Model-based Reinforcement Learning." Learning for Dynamics and Control. PMLR, 2020.
- **experimental quality**: the numerical results are impressive, but I don't see enough explanation or intuitions for them. Especially when continuing to browse the ablations in the appendix, I think the experiments need to be presented with more understanding for the results to be reproduced.
- **writing quality**: the writing of this paper could be substantially improved. The writing has many typos and the structure is not very clear.
- In a couple reads I never understood the part on the directional derivative as a scalar. Can the authors help me understand this? I acknowledge this may be common background that I am missing, so it may be a useful addition to the appendix.
- If combining the proofs with DDPPO is too difficult, maybe this should be two separate papers.

## Comments By Section(s)

### 1 Introduction
- This section was a bit too broad, which made it hard to get the point of the paper across. The author's do not need to describe everything about RL here or the history of MBRL (related works do that, and end up being repetitive)
- This sentence is crucial, but both worded weirdly and not emphasized enough in the section "Note that there is a big difference between these methods ... when calculating the policy gradient in these methods"
- The paper could do better at clarifying the contributions of the mathematics and the algorithm. The authors discuss a method and then a new algorithm. Maybe this can be combined into one point, but it is hard to follow and see where the authors are going. The contributions help this, but by that point there are more questions than answers.
- the second half is very repetitive. Trimming this will make the paper easier to follow and add space to other sections that need it.

### 2 Preliminaries
- The paragraph introducing equation 4 is confusing. If it is a "preliminary", which paper should I look at to know more? Please cite that so I can learn it / follow the paper.

### 3 Related Works
- This paper could be good to include in the related works. Kégl, Balázs, Gabriel Hurtado, and Albert Thomas. "Model-based micro-data reinforcement learning: what are the crucial model properties and which model to choose?." ICLR 2021.
- the formatting here is really forced and deteriorates the reading experience.
- The related works seem dismissive of the past work. It is clear this paper is different, but why are past works important?

### 4 Model-based Policy Optimization by Considering the Model Gradient Error

#### 4.1 CONVERGENCE RATE FOR MODEL-BASED POLICY OPTIMIZATION
- in the intro, this sentence makes me think there will be a general explanation of how this can be used for algorithms, but it seems to be lacking "for the policy optimization algorithms in what the policy gradient is calculated using the learned environment model"
- This section needs substantially more explanation to support the math. Some assumptions can be moved to the appendix unless it is explained in the text why they are crucial and new.
- Are there related works that hint to these formulations? They are big step and hard for me to follow. Knowing this, the authors should spend more time making them "land".
- I think there is a typo in the remark after theorem 2 -- the learning rate alpha_t should be 1/sqrt(T) rather than sqrt(t)?

#### 4.2 TWO-MODEL-BASED LEARNING POLICY OPTIMIZATION
- very repetitive. Can remove some here or in intro.

#### 4.3 DIRECTIONAL DERIVATIVE PROJECTION POLICY OPTIMIZATION: A PRACTICAL IMPLEMENTATION
- I believe fairly strongly the algorithm should be in the main body of the paper, as it is sufficiently different from past. work.
- Otherwise, a lot of this section is again repetitive. Please separate the repeated parts of the paper to where they should be. Intro is for framing, preliminaries for background / common math, and the new stuff in your method :). For example, training a model with MSE / MLE is described multiple times and can be done with only a citation.
- this sentence is interesting and potentially crucial to the result "So in this paper, we sample the nearest n data points of x to calculate n directional derivatives in n different directions and use the finite difference to co". It would be interesting to hear more intuition on why this works.
- Figure 1 says "trials" - how many each, and how many for other algorithms?
- A system diagram could make it clearer how the algorithm works.

### 5 Experiments
- Figure 2 seems to be in too simple of environments to validate the claims totally. The strong performance of humanoid is a big piece of the results, how do these intuitions translate to higher dimensional tasks?
- What happens if the gradient model is used for state rollouts? This could be an interesting way to address the objective mismatch issue.
- which implementation is used for the compared algorithms?

- Why were Hopper and Ant chosen for Figure 3? Were these sweeps tried on other environments? Are the results consistent. At least commenting on these lines and or why the trends would break down are important to people considering using the method.
- Hard to tell if the two model part is needed by the results? The impressive humanoid results are diluted by me when considering it's not the key variables the author's propose contributing to success. Why is there little gain from the gradient-trained model when adding the accuracy model? The labels on figure 4/5 could be more precise. One model is not the most clear.

----

## Grammatical comments / nits
- abstract "We" -> we
- intro weird phrasings "decision-making problem is always formulated as", "agent uses policy to determine" -> determines, "the rollout-based model-based methods that the model gradient is crucial" -> where the model gradient is..., "we can see that the transition and reward model gradient matter in"
- intro confusing "Therefore, we propose to separate different requirements for the model into different models"
- "w.r.t" missing period before eqn. 3
- "the expectation is taking over" -> "taken over" after eqn. 3
- related works "policy gradient to re-weighted... re-weighted" weird, also missing period at the end of this sentence ... "sample Use the learned"
- section 4 "that using the policy gradient" that are using?
- should use latex references for section names, referred to "Preliminary section" many times with different capitalization.
- wrong " in 4.1 "tilde" notation, "hat"
- theorem 2 remark: "gradient method converge to its..." -> converges
- in 4.2 "between collecting samples from the environment, training the model, and optimize the policy" --> optimizing
- in 4.3 "Hence in this paper, we use the error between the estimated directional derivative and the projection value to constraint the learned model’s gradient" --> constrain
- "Model Usage" section: "we use the objective function that constructed by following" --> is constructed, " , " space before comma later

----
## Other comments / ideas (not included in scoring of paper)
- in the appendix, the authors include hyper parameters. Do you all have an intuition for why the higher dimensional tasks need fewer neighbor points, H, to calculate the derivative? As the density drops, I thought more points may be needed?
- There is a really interesting correlation in figures 4 and 5 in the appendix that is not explained, but could be a great intuition to support the paper. The environments where the one-model approach fails to perform as well seem like they could have a bigger difference in the model errors? I am looking at the Hopper env in figure 4 and 5. Though, the ant does not back this up as much.


**Summary Of The Paper:**

The authors present a new approach for model-based reinforcement learning by training two models and using them separately. One model prioritizes accurate transitions and another has a weighting term for value-gradient, which they combine in a new algorithm directional derivative projection policy optimization.


**Summary Of The Review:**

The paper proposes a novel solution to a well known problem in model-based RL. Though, the paper is not well written so it is hard to follow, even for an expert in the area. In its current state, I do not recommend acceptance because of its writing / presentation / lack of conveyed intuition, but these are problems that can be fixed in a moderate revision.

---

> ### Author Response · Authors · 2021-11-18
> **To Reviewer 8zAV (Part I)**
>
> We are sincerely thank the reviewer to recognize our motivation and novelty. Thank you very much for your detailed comments and suggestions.  We provide our responses to your review comments.
>
> **1.	General comments**
> -	We do not know the paper you provided before. After reading it, we find that this paper does not relate to the importance of gradient information when learning the model. However, the idea about considering the “mismatch” at model learning is insightful and can help to understand our job. We have added it to the introduction part.
> -	We revise our paper and add a more comprehensive explanation for the experiment part in both the main body and the appendix.
> -	We fixed the typo in the paper by proofreading it multiple times.
> -	The directional derivative can be viewed as the projection length of the gradient vector into one certain direction. We add detailed explanations into the appendix B.
> -	We revised the proof section in the appendix to make it clearer and more intuitive.
> **2.	Comments on the Introduction section**
> -	We have moved some basic information about RL or the history of MBRL to related work and have added words to explain the crucial sentence the reviewer mentioned.
> **3.	Comments on the Preliminaries section**
> -	Intuitively, Eq. 4. describes the policy update step using approximated policy gradient. It is a commonly used gradient accent or decent step doing gradient-based optimization. We add the descriptions into the manuscript.
> **4.	Comments on the Related work sections**
> -	The past work is indeed important. Our method builds on the past work that considers the model prediction error. What we want to emphasize is that previous works missed something important which motivate our paper.  We revise our related work section to make this point clear.
> -	We revise our related work section to make it easier to read.
> -	We do not know the paper you provided before. After reading it, we think this paper is not directly related to our job. However, this paper is important in model-based RL area that firstly give a measure to tell which model to choose, so we have added it to related work in Appendix A.
> **5.	Comments on the Section 4.1**
> -	We move the unnecessary assumptions into the appendix to make this section clearer.
> -	We add more explanation to make this section easy to understand. Actually, in the proof of Theorem 2, we need to prove the convergence rate of a non-convex optimization problem with a biased gradient. The most challenge part is to bound the bias brought by the inaccurate model and the non-i.i.d. data. We provide three papers and explain the main idea of the proof before the proof. We hope this will help to understand the theoretical results.
> -	This is not a typo. We choose to set a changing learning rate as commonly used. So, the learning rate will be $\frac{1}{\sqrt{t}}$. However, choosing a fixed learning rate as $\frac{1}{\sqrt{T}}$ is also feasible.
> **6.	Comments on the Section 4.2**
> -	Thanks for your suggestions about our paper structure. We are more than happy to polish our paper according to your suggestions.
> **7.	Comments on the Section 4.3**
> -	We move the algorithm framework to the main body according to your suggestion
> -	We carefully read the paper and adjust the paper structure to make it clearer and more informative.
> -	As shown in the paper, we don’t know the ground-truth gradient when facing a real reinforcement learning task. So, we need to estimate the gradient vector by using data. We can use the nearest n data to estimate n directional derivatives and using these directional derivatives we can reconstruct the gradient vector. We detailly explain it in the new version of the manuscript.
> -	We already say that there are five trials in the second line of the title of figure1. For other algorithms, we use five too. Five trials are a commonly used random seed number in many of the baseline papers considering the limitation of the computing resource.

---

> ### Author Response · Authors · 2021-11-18
> **To Reviewer 8zAV (Part II)**
>
> **8.	Comments on the section 5**
> -	Please note that two experiments in Figure 2 are select because we can analytically calculate the gradient of them. Although these two experiments are simple, the algorithm is the same as high dimension tasks. Therefore, we leverage these experiments to clearly demonstrate our motivation.  For higher dimensional tasks, it is unable to get the true gradient on Mujoco, so we can only see the overall performance on it.
> -	We think the results shown in Figure 5 (Figure 6 in revised version) in the appendix can address your problem. It shows that the prediction error of the gradient model is larger than the original model which means using the gradient model to rollout may cause worse results. Meanwhile,  the curves labeled "one model(“single model” in revised version)" in Figure 3 perform worse which are consistent with this view.
> -	For MBPO, we directly use the number provided by [Michael Janner](https://github.com/JannerM/mbpo);  For SAC we use the code and hyperparameters from [github link](https://github.com/pranz24/pytorch-soft-actor-critic); For MAAC, since there is no open-source code available, we reproduced it by ourself. The model structure and hyperparameters such as learning rate are the same with MBPO, and the extra hyperparameter, MVE length, we set 2 for hopper and 3 for others which is consistent with ours. For other baselines, we use the codes or results from [Model Based Reinforcement Learning Benchmarking Library](https://github.com/WilsonWangTHU/mbbl). For a better comparison, we have additionally added the details of the baselines to the appendix in our revised version.
> -	For the ablation study, we desire to use a simple env (low dimension) and a complex env (high dimension) to show the performance, so we just present the result from Hopper and Ant. Please note that for all six experiments, we fix all the hyper-parameters not related to our contribution the same as the baseline method. We search the hyper-parameters fairly as we present in the Appendix B.2.
> -	The two-model part is important, and the conclusion is supported by our ablation study.  As shown in the Figure 5 in the revised version, the performance of "two model"  on Hopper, Halfcheetah, Ant, and Humanoid is better compared to the "single model (“one model” in old version)" structure. “ single model”  in revised version means we only train a single model using both prediction error and gradient error as supervision.
> **9.	Other comments**
> -	We just try the different number of points on Hopper and Ant, the results tell us to set a relatively small number for higher dim env. We think this is a trade-off, if there are enough points in the buffer, then it is better to use more points to calculate the gradient. However, if there are not enough points and we still sample too much, the distance between the sampled point and the target point, where the gradient should be calculated, will be large. In other words, we not only hope there are enough points, but the sampled points are close enough to the target. So it may be a trade-off and explains why a small number of points is better for Ant.
> -	The fails of "one model" may be caused by the worse rollout due to a large predictive error. The difference in figure 5 (figure 6 in revised version) for ant is not obvious as Hopper, we don't understand it yet, maybe it's caused by the nature of the environment.

---

> ### Comment · Reviewer_8zAV · 2021-11-19
> **response to authors**
>
> Thanks for the detailed responses. I have looked at the new version of the paper, and it is definitely better. I am still borderline in my recommendation. The empirical results show better sample efficiency on high-dimensional tasks, but why is this?  I am not surprised to see it, but I don't think this results of sample efficiency is supported elsewhere.
>
> The details added to the text and appendix do clarify the experiments. You all did an impressive amount of comparisons and collection, which I certainly appreciate.
>
> Also, the changes to the proofs represent a lot of progress to making the paper more readable.

---

> > ### Author Response · Authors · 2021-11-24
> > **Thanks for your positive feedback. A short discussion for an interesting problem.**
> >
> > Thank you for your recognition of our revised paper. Your detailed comments help us improve the quality of the paper significantly.
> >
> > It is interesting to think deeply about the relationship between performance and the difficulty of the problem. We think your further questions raise an interesting future work for us, and we are working on the theoretical justification for this problem.  We are happy to share our results after we finish it.
> >
> > Here, we show some intuition in our minds now. We hope that these insights may address your questions partially.
> > We think the performance on high-dimensional environments improves a lot may because the true models here are more “complex” (e.g. larger Lipschitz coefficient, higher dimension). In the complex model, we are more likely to make mistakes in the gradient direction but keep the prediction error small.
> > Since our model can explicitly model and learn the gradient direction, the accuracy of the gradient predicted by the model may improve larger on 'complex' problems than the counterpart.
> >
> >
> >
> > We design a synthetic experiment to see how the “complexity” influences the gradient error of models with and without gradient loss. In the experiment, we choose a function $f(x)=sin( a \sum_{i=1}^d sin(x_i) )$ with d-dimension input and 1-dimension output. Here a higher $a$ and higher dimension means a more complex model to learn. We then use these data to train a model with both prediction error loss and gradient error loss. As a comparison, we also train a model with only prediction error loss. We show the improvement of the accuracy of the calculated gradient by using our method. The synthetic experiment results are shown in the following:
> >
> >
> >
> > |  Train |      a=2   |         |          |      a=8   |         |          |
> > |:------:|:--------:|:--------:|:--------:|:--------:|:--------:|:--------:|
> > |        |    w/o grad loss  |     w/ grad loss   |  improvment |    w/o grad loss  |     w/ grad loss   |  improvment |
> > |  dim=5 | 0.002217 |  0.00067 | 0.001547 | 2.719887 | 0.168487 |   2.5514   |
> > | dim=10 | 0.005417 | 0.002361 | 0.003057 | 8.446585 | 1.063909 | **7.382675** |
> >
> > |  Test  |    a=2   |     |          |    a=8   |      |          |
> > |:------:|:--------:|:--------:|:--------:|:--------:|:--------:|:--------:|
> > |        |    w/o grad loss  |     w/ grad loss    |  improvment |    w/o grad loss  |     w/ grad loss    |  improvment |
> > |  dim=5 | 0.005194 | 0.001158 | 0.004036 | 5.404485 | 2.487458 |  2.917027  |
> > | dim=10 | 0.004503 | 0.002663 | 0.00184  | 15.34161 | 8.508946 | **6.832661** |
> >
> > By comparing the improvements with different $a$ and different dim, we find a complex environment with larger $a$ and higher dim can cause higher improvements which probably be the reason why our method performs better on these high-dim problems.

---

### Official Review · Reviewer_nkhB · 2021-11-01

**Correctness:** 3
**Technical Novelty And Significance:** 4
**Empirical Novelty And Significance:** 3
**Recommendation:** 8
**Confidence:** 3

**Main Review:**

I am recommending to accept this paper because its algorithm construction is well-reasoned, the theory appears to be novel and provides appropriate support for the proposed algorithm, and the empirical results provide some minor support for the claims. Overall, I find that this paper contributes an important perspective on the role of using differentiable models, and the need to control the error of first-order information (gradient model) instead of only zeroth-order information (prediction model) when using differentiable models.

I remain on the fence, however, because (W1) the empirical results provide only marginal support for the proposed algorithm and (W2) there is a disconnect between the paper's stated goals and the novelty the paper _actually_ provides, leading to neither being done particularly well.

---

**Details for W1:**

 While the proposed algorithm is the only algorithm to consistently perform well across all 6 benchmark domains, the paper does not provide sufficient information about the baselines to understand if this experiment is fair. A couple of immediate concerns come to mind:

1. How are the hyperparameters selected for the baseline algorithms?
2. Because some baselines are model-based, some are model-free, and many baselines use a different number of models, is the comparison between baselines fair in terms of compute usage and number of free parameters? The proposed algorithm uses an additional model to encode the world model gradient, suggesting that it has more learnable parameters than any other baseline. Do these extra parameters account for the performance difference alone, or is the performance difference actually due to decreased bias in the policy gradient?

The use of only 5 random seeds is also concerning, especially considering the very high variance exhibited in many of the reported results. Are any of these results statistically significant? Do any statistical significance tests allow for the use of 5 samples? Does this imply that the reported results could simply be due to random chance?

**Details for W2:**

Largely due to the empirical concerns stated above, it seems very challenging to support the claim that the proposed algorithm achieves state-of-the-art results. Consider additionally that only a small subset of the standard Mujoco domains are used significantly limits the scope of the SOTA claims that can be made. Proposing a novel algorithm which uses another NN-worth of free parameters over baselines, then claiming SOTA does beg the question whether SOTA was achieved only through more parameters.

However, I think the real value provided by this paper is further insight into the need to control the gradient error when using differentiable models. The paper performs two interesting ablation studies, one showing that the proposed algorithm does successfully control this error term, and the other showing the impact of controlling this error on the overall performance. However, neither of these ablations are complete; especially the second.

Figure 3 shows that adding the gradient loss term to control this bias has negligible impact on the performance of the proposed algorithm across all three manipulations (number of directional derivatives sampled, different weightings for the added loss term, and different methods of controlling the gradient error). Considering further that two of the six domains were cherry-picked for reporting in the main body of the paper, suggests a post-hoc maximization bias occurred while selecting these results as well. In order to well-support the theory proposed in this paper, I believe the most important set of experiments is actually the ablations. These should be significantly expanded and further insights should be drawn about the role of controlling the gradient error term. In order to make room for this expansion, the SOTA experiments can be largely treated as a demonstration that the proposed method scales well and is competitive, with considerably less focus placed on Figure 1.

**Summary Of The Paper:**

This paper seeks to improve performance of model-based policy optimization methods by taking advantage of the world-model's differentiability. It starts by showing theoretically that error in estimating the model's gradient contributes to bias in the learned policy. The paper then proposes directly optimizing the jacobian of the world model using a sample-based strategy in order to directly control this bias term.

**Summary Of The Review:**

**Edit during discussion phase:**

I have increased my overall score from 6 -> 8 based on the ongoing discussion. I have also increased my "empirical novelty" score from 2 -> 3 because I believe the new ablations and clarifications yield deeper insight into the impact of using first-order information to constrain transition models in RL.

---

(Copied from above Main Review)

I am recommending to accept this paper because its algorithm construction is well-reasoned, the theory appears to be novel and provides appropriate support for the proposed algorithm, and the empirical results provide some minor support for the claims. Overall, I find that this paper contributes an important perspective on the role of using differentiable models, and the need to control the error of first-order information (gradient model) instead of only zeroth-order information (prediction model) when using differentiable models.

I remain on the fence, however, because (W1) the empirical results provide only marginal support for the proposed algorithm and (W2) there is a disconnect between the paper's stated goals and the novelty the paper _actually_ provides, leading to neither being done particularly well.

---

> ### Author Response · Authors · 2021-11-18
> **To Reviewer nkhB**
>
> Thank you for your positive comments on our work. As you are saying, using differentiable models is a totally different scenario compared to the traditional classification and regression tasks, and it requires more information when learning the model. We provide our response to your concerns below, and we hope this will reassure you.
>
> **1.	How are the hyperparameters selected for the baseline algorithms**
> - For MBPO, we directly use the number provided by [Michael Janner](https://github.com/JannerM/mbpo);  For SAC we use the code and hyperparameters from [github link](https://github.com/pranz24/pytorch-soft-actor-critic); For MAAC, since there is no open-source code available, we reproduced it by ourselves. The model structure and hyperparameters such as learning rate are the same with MBPO, and the extra hyperparameter, MVE length, we try different steps and finally set 2 for hopper and 3 for others which is consistent with ours. For other baselines, we use the codes or results from [Model Based Reinforcement Learning Benchmarking Library](https://github.com/WilsonWangTHU/mbbl). For a better comparison, we have additionally added the details of the baselines to appendix B.2 in our revised version.
>
> **2.	Do these extra parameters account for the performance difference**
> -	The answer is NO.
> -	To address the concern, we additionally add experiments in our revised version that the number of parameters used by the two models in total is the same as the one used by the baseline with a model alone. The lines labeled `baseline(#param ~2M)` in ablation  mean learning the model without gradient information but using a bigger number of parameters. From the results, we can see that directly added parameters don't make much improvement which yields the value of our method. Besides these, we will add experiments that all baselines using a model with a bigger size when finished.
>
> **3. 	5 random seeds**
> - Our implementation is mainly based on MBPO, and we just use the same number of trials (5) as [MBPO](https://arxiv.org/pdf/1906.08253.pdf). To address your concern, we use a t-test to test the hypothesis that is whether the mean performance of our method is significantly larger than the baseline method (MAAC). Due to the time limit, we will add 5 more seeds to each experiment into the camera-ready version.
> The results of t-test is(p value < 0.05 is a commonly used threshold to show the significance):
>
> |env name|Hopper|Walker2d|Halfcheetah|Ant|Humanoid|
> |---|---|---|---|---|---|
> |env step|75k|75k|75k|100k|100k|
> |p value|0.0003|0.0389|0.0139|0.0369|0.0015|
> |env step|100k|100k|100k|150k|150k|
> |p value|0.0004|0.2967|0.0015|5.3e-5|3.0e-7|
>
> **4. 	Achieve SOTA and parameter numbers**
> - What we want to emphasize is that our algorithms can achieve better performance using fewer samples. The claim “SOTA” is confusing, and we replace it with a more accurate term “better sample complexity”.
>
> **5.	Completeness of the ablation study**
> -	Please note that two experiments in Figure 2 are selected because we can analytically calculate the gradient of them.   We further add experiments to show the impact of controlling this error on the overall performance. The results are shown in Figure 2.
> -  For higher dimensional tasks, it is unable to get the true gradient on Mujoco, so we can only see the overall performance on it.
>
> **6.	Concern about our ablation study**
> -	First of all, we do not cherry-pick experiments for reporting. We select Hopper(low dimension) and Ant(high dimension) according to their dimension. The first two ablation study (number of directional derivatives sampled, different weightings for the added loss term) aims to show that our algorithm is robust to the extra hyper-parameters tuning in a reasonable region. The third ablation study aims to show the importance of the two-model structure.  We put all six domain results of the third ablation study in the appendix (Figure 3)
> -	To address your concern, we add one more ablation study together with more comprehensive words to explain the results of the ablation study.

---

> > ### Comment · Reviewer_nkhB · 2021-11-22
> > **Response**
> >
> > Thank you for the detailed response! I am happy with the changes for (1, 2, 4, and 5) above.
> >
> > For **(3)**: this still isn't a sound experiment. There is no way to check that the assumptions for the t-test are valid using only 5 samples, so these p-values are meaningless. The proper way to assert statistical significance would be to first check the shape of the distribution (often this step alone requires ~20 samples as a rule of thumb), then pick an appropriate test. Bootstrap-based tests are common given that performance distributions in RL tend to be long-tailed or bimodal.
> >
> > For **(6)**: this mostly addresses my concern, thank you. I would additionally suggest noting in the text why you selected the given environments since it appears there was some prior information influencing this choice (which is a good thing, using posterior information would be bad).
> >
> > ---
> >
> > Considering that many of my primary concerns were addressed, and after having read the other reviews/discussions, I am increasing my score and will be advocating for acceptance of this paper.

---

> > > ### Author Response · Authors · 2021-11-23
> > > **Thanks for your positive feedback on the updated paper**
> > >
> > > We are more than happy to know that our response can address your two concerns.  Your valuable comments help to improve our work.
> > >
> > > For (3), your comments are correct. Although 5 seeds are always adopted, it is far from sound to test the significance of the experiment results. We are continuously working on more random seeds to make our work more sound. Due to the limited computation resource, we will share the results when we finish it.
> > >
> > > For (6), thank you for your great suggestions. In fact, we had falsely thought that we already clarified the reason for the environment’s selection in the ablation study. But for some reason, we miss these words. We will add it to the main body of our paper.

---

### Official Review · Reviewer_TLT1 · 2021-11-03

**Correctness:** 3
**Technical Novelty And Significance:** 2
**Empirical Novelty And Significance:** 2
**Recommendation:** 6
**Confidence:** 4

**Main Review:**

I like the insight that method can use the model's predictions and gradients and thus we should be aware of model gradient errors. However in practice, we do not know the gradients on the trajectories that we attain and equation (12) requires computing the n-nearest points of $x$ in the replay buffer to estimate the model gradient. This seems potentially intractable in practice for large replay buffers, and seems like it will not always coincide with the true model gradient if the buffer is sparsely filled. Is the scalability here the reason why the methods were not run for more timesteps?

The experimental results in Figure 1 show the sample efficiency and show an improvement over the relevant baselines, using 5 trials. The most relevant baseline here is to MAAC, as Eq 14 is set up very similarly. I especially like the ablation in Figure 3(right)/Figure 4 that ablates against the model not using the separate gradients.

One slight concern I have here is that **some of the empirical gain may come from modifying the base model-based algorithm rather than adding the gradient information.** I see the value being optimized in Eq 14 as nearly identical to the one optimized in MAAC (S4.1 there), but some of the baseline results (without the gradient loss) in Figure 4 are better than the MAAC baseline in Figure 1 (especially the cheetah/humanoid).

My largest concern with the paper is that the end of Section 5.1 says that the results of ~5k reward attain "near-optimal policies in the Humanoid environment" (for running) despite papers such as the [SAC-SVG](https://arxiv.org/abs/2008.12775) attaining a reward of ~9-10k in the same environment. This paper does not provide a video of the learned policy, and furthermore, it is possible for the humanoid agent to remain stationary (i.e. standing still) the entire episode and attain a reward of 5k (because the [base alive bonus is 5 reward/timestep](https://github.com/openai/gym/blob/bb81e141ea7ae67ce339109095841592e8231185/gym/envs/mujoco/humanoid.py#L34) and the episode runs for a maximum length of 1000 timesteps.) **I think this is extremely misleading and should not be accepted for publication.** I request for the authors to clarify this point and provide a video of the agent learned on this task. I am very willing to further discuss this point and increase my score on this paper if this is properly addressed.

**Summary Of The Paper:**

This paper considers learning a model for reinforcement learning and proposes to also learn the gradients of the model in addition to just the predictions. They approximate the model gradient using the nearest samples from the replay buffer in (12) to learn the model gradient (13), which they use to replace the model gradients in the value expansion in (14). They evaluate on MuJoCo environments (Fig 1) and show some key ablations (Fig 3) on the number of samples used for the gradient estimates, and the weight term in (12), and combined/separate models.

**Summary Of The Review:**

This is an insightful paper on modeling the gradients that contains an extensive empirical evaluation. My main concern is in the presentation of the humanoid results.

---

> ### Author Response · Authors · 2021-11-18
> **To Reviewer TLT1**
>
> Thanks for your recognition of our paper’s insight. Also, thank you very much for the valuable comments. I think most of your concern is misled by our confusable presentation which is fixed in the revised version of the manuscript. Here are our responses to your 3 concerns.
>
> **1. Your largest concern about the presentation of the humanoid**
> -	Please note that the main goal of our proposed method is sample complexity. The term “near-optimal policy” in the original paper is inaccurate and what we actually want to emphasize is that we can achieve better or comparable performance using less sample compared to the baseline model-based and model-free methods.  We revised our paper in the abstract, introduction, and experiment sections to clarify this point.
> -	The paper SAC-SVG does not target the sample complexity, and it uses 5e6 interactions to achieve the “near-optimal”.  To obtain around 5k return, as most of the baseline achieved, SAC-SVG uses about 1e6 interactions which is extremely larger than our method (1e5 interactions). Considering the SAC-SVG shows big potential in the asymptotic performance, we believe combining our method with the SAC-SVG is an interesting and potentially efficient algorithm.
> -	We find that the policy learned by baseline methods (MAAC, SAC) and our algorithms let the humanoid agent remain stationary. Per your requirements, we provide the videos and open source of codes in [github]( https://github.com/paperddppo/ddppo/tree/main/ddppo). We think that increasing the asymptotic performance of the policy optimization algorithms especially escaping the local optimal is very important, but it is not the main problem that our paper aims to solve.
>
> **2.	Your slight concern about the empirical gains**
> -	We believe the adding gradient information contribute to most of the experiment’s gains. Meanwhile, our implementation of model-based algorithm is almost the same as MBPO except for the gradient model learning and the policy gradient calculating which are related to our main contribution.
> -	On the one hand, we use the simulation experiments (Figure 2) to demonstrate the effectiveness of using gradient information can effectively reducing the gradient error in the simple environment.
> -	On the other hand, in our implementation, we follow the methods used in MBPO that rollout for $k$ steps on the learned model to collect fake data for Q updating. Please note only combining existing model-based algorithms does not provide a consistent guarantee for performance. For example, in the Ant experiment, our implementation without gradient information is worse than MAAC but increases a lot when adding gradient information.
>
> **3.	Your concern about the efficiency of computing the n-nearest points in the replay buffer**
> -	In practice, we use KD-tree to construct and search the top-n nearest points in the buffer for each point. The averaged search and insert complexity are both $O (log_2 N) $.
> -	The model learning procedure is the same as MBPO [1] that trains the model every 250 env-steps and uses early stop. We just need to search $n$ nearest points for each point in the buffer before the model training begins. Specifically, we first sample $min(100000, buffer\ size)$ points, and then for each point in the buffer, we search the $n$ nearest points in this sampled dataset. So, the complexity will not go explosion.
> -	The running time for building KD-tree and searching n nearest points one time is around 7.7s for 50k samples and 8.32s for 100k samples in hopper(dim=14). Please note that we only need to do building and searching 4 times in 1k environment step.

---

> > ### Comment · Reviewer_8zAV · 2021-11-19
> > **Humanoid**
> >
> > Just wanted to chime in looking at the videos there is very little substance there to warrant a decision either way. The policy for DDPPO is not compelling, but neither are the other baselines. Therefore, the numerical difference is somewhat of a red herring in my opinion.
> >
> > In general, model-based methods struggle on humanoid and have not solved them impressively (from states), so I think this result fits with the literature.

---

> > ### Comment · Reviewer_TLT1 · 2021-11-20
> > **Raising score from strong reject to weak accept**
> >
> > Thank you for editing all of this in your paper and sending the code/videos of the agents. These address my biggest concern on the presentation of the humanoid results and I've updated my score from a strong reject to a weak accept. I lean towards accept as the idea and modeling insights seem valid and interesting. But, I still do not find the experimental gains significant or interesting, especially on the gym humanoid where it's only faster-learning to find a poor local optimum of the gym humanoid.

---

> > > ### Author Response · Authors · 2021-11-23
> > > **Thanks for raising the score**
> > >
> > > We are very glad that our responses can address your biggest concern on the presentation of the humanoid results. Thank you for your recognition of the idea and modeling insights of our work and thanks for raising score.
> > >
> > > Also, thank you for pointing out the local optimal problem which is very important for the model-based RL. But we disagree with the opinion that our proposed algorithm can only find a local optimum of the gym humanoid.
> > > Our reasons are listed as follows:
> > > - Our proposed algorithm mainly targets how to learn a “better” model. Our method can be thought of as a method for learning the model so that can be embedded with other basic methods. Therefore, the proposed method is general enough to combine with other model-based RL algorithms.
> > > - We pay more attention to the sample complexity while achieving a comparable performance as “base” model-based RL algorithms. In this work, our base model-based RL algorithms are mostly related to MBPO, MAAC, and SAC.  And we indeed demonstrate our ability to improve the sample complexity.
> > > - It is of great interest to combine our method with other methods that have good asymptotic performance such as SAC-SVG. Since combing our method with SAC-SVG are easy and without fundamental challenge, we are combining them together and are glad to share the results when finished.

---

> > > > ### Comment · Reviewer_TLT1 · 2021-11-23
> > > > **It will be great to see further developments and extensions!**
> > > >
> > > > Indeed I agree separate gradient models can help in settings beyond those shown in this paper and it's worth investigating and scaling. The current state of the experimental results is **objectively** that the "proposed algorithm can only find a local optimum of the gym humanoid". (In case this comes from an ambiguity in the language we're using -- of course, there are other experimental results, but on the gym humanoid, the policy is objectively only a local optimum.) I could believe that this model will help go beyond this local optimum but it needs to be empirically demonstrated.

---

> > > > > ### Author Response · Authors · 2021-11-24
> > > > > **Thank you for your great suggestions.**
> > > > >
> > > > > - Thanks for your response. We really appreciate that you discuss with us and recognize our idea and insights. Your comments and suggestions help us much when preparing the revised version of the manuscript.
> > > > > -  We think your suggestions point a great direction for future work. We are happy to test our method in a wide range of environments and algorithms.
> > > > > - Please feel free to discuss if there is anything else to clarify.

---

### Author Response · Authors · 2021-11-23
**General Comment and Updated Manuscript**

We thank all reviewers for their constructive comments and suggestions which help to improve the manuscript.
Also, we thank for the reviewers’ recognition of the idea and insights of our work and for being willing to raise the score. We summarize our main revision of the manuscript and new materials below.

- According to the reviewer’s suggestions, we provide the code for implementation of our method on [github]( https://github.com/paperddppo/ddppo/tree/main/ddppo).   We also provide videos of the performance of our agent on humanoid.   Hope this can help understand the algorithm proposed by our work.

- Thank all reviewers for their suggestions on the writing of our paper. The introduction section is streamlined. And we add a more intuitive explanation to make it easy for reading. Citations are added to make the preliminaries easy to follow.  Two more citations are added to make the related work more complete, and we rewrite the words to make it easy for reading. For section 4, we summarize the necessary assumption and put it before the theorem. We move the unnecessary assumptions into the appendix to make this section clearer per reviewers’ suggestions.   We remove some repetitive words in section 4 to make room for moving the pseudo-code from the appendix to here. We add some content to explain the loss function of the gradient model and why the gradient loss alone would be insufficient for training a model.  For section 5, we adjust the legend to make the figures clearer in the ablation part. For the appendix, we add a sub-section to introduce the details of the baselines and our implementations. We add two sub-sections in the appendix to introduce some basic information for a better understanding of our work. In the part of the proof of the main theorem in the appendix, we add some explanatory words to make the proof easier to follow.
- According to the reviewer’s suggestions, we add two more experiments. The first is to make section 5.2 complete. We draw the overall performance of our method and the performance without consideration of the gradient information in Figure2 to support our result. The second one is to address the concern that the performance improves just because of the increased parameters. We add an ablation study to compare the performance of our method and the baseline with a similar number of parameters which yields the importance of gradient loss.

- Last but not least, we thank all reviewers again for the effort they put into this which makes this paper more readable and well-structured.

---

### Public Comment · ~Zhang_Minghao1 · 2023-11-23
**Seeking Clarification on Performance Differences in 'Gradient Information Matters in Policy Optimization by Back-propagating through Model**

I have a few points of concern regarding this paper:

1，In my own attempts to reproduce the results, particularly on the Pendulum task, I did not observe a significant difference in performance between using gradient loss and not using it. The improvement was less than 2%, which is markedly different from the 50% improvement reported in the paper. I am curious as to why this might be the case.

2，When comparing to state-of-the-art results, I found that this method did not outperform the MBPO algorithm based on DI-engine (https://di-engine-docs.readthedocs.io/en/latest/12_policies/mbpo.html). I was wondering if there could be a specific reason for this.

---

> ### Public Comment · ~Chongchong_Li1 · 2023-11-23
> **Clarification**
>
> Thank you for your interest in our paper.
> 1. Our results show how the performance of the agent on pendulum changes with the increase of the number of interactions with the environment. The paper did not conclude that there was a 50% improvement. Based on the performance curve, we can only see the score that can be achieved with the same number of interactions. As for your experiment, we do not know your specific settings, here we suggest lr=0.0003, h=4, weight=1.0, near_n=5, which are also the parameters used in our paper.
>
> 2. For MBPO, we directly use the reported number given by the author Janner et al. (https://github.com/JannerM/mbpo). To reproduce  our results, please use the docker (image: xfdywy/work:py37pt150mujocogpu) to ensure consistency of the environment. Additionally, different mujoco environments have their own hyperparameters, which are recommended to be set as described in Section B.2 of the paper. Of course, we can provide the rest of the bash files to run the program if you want, or simply provide checkpoints of the trained models to explore further. If you have any further questions, please feel free to contact me (18118002@bjtu.edu.cn) and we are more than happy to have a further discussion.

---

### Decision · Program_Chairs · 2022-01-20

**Decision:**

Accept (Poster)

**Comment:**

The paper considers model-based RL, and focuses on approaches that benefit from the differentiability of the model in order to compute the policy gradient. It theoretically shows that the error in the gradient of the model w.r.t. its input appears in an upper bound of the error in the policy gradient computing using the learned model. Motivated by this, it suggests a MBRL approach that learns two models, one of them minimizes the next-state prediction error (as commonly done) and the other minimizes a combination of prediction error and the gradient error.
The paper empirically studies the method through extensive experiments.

Reviewers are generally positive about this work. They believe that the paper is insightful and the method is original. At first, there were some important concerns raised by the reviewers, but the authors revised their paper in the discussion period, and it appears that the reviewers are all satisfied now. I also read the paper during the rebuttal phase, and I should say that I have some concerns myself, especially on the theory part of the paper. Given that the authors did not have an opportunity to answer my questions, I do not put much weight on my concerns (and I believe most of them can be addressed with some clarifications). Considering the positive response of reviewers and promising results, I am going to recommend **acceptance** of this paper.

I strongly encourage the authors to consider the comments by reviewers, as well as the following ones, in the revision of their paper.


**Comments**

1) The true dynamics $f$ is defined as a stochastic one, i.e., $s_{t+1} = f(s_t, a_t, \epsilon_t)$ (just before Eq. 1), and similarly for the learned model. Here $\epsilon_t$ is the noise causing the stochasticity of the model. But later, when the errors on the model and its gradient are introduced (i.e., $\epsilon_f$ and $\epsilon_f^g$), the role of stochasticity becomes unclear.
For example, we have
$\|| \tilde{f}(s,a) - f(s,a) \||  \leq \epsilon_f$.

What happened to the noise term?

The same is true for Eq. (5). The next-state s' (either according to the true dynamics or the learned model) is random. In that case, it is not obvious how to interpret Eq. (5). Is it the error of the expected gradient of the next state? Or is it something else?

In case the dynamics is assumed to be deterministic, this should be clarified early in the paper.

2) The upper bound in Theorem 1 might be vacuous if the Lipschitz constant $L_f$ of the model is larger than 1.
To see this, consider Lemma 1. The constant $C_0$ is $\min [D/\epsilon_f, (1-L_f^{t+1})/(1 - L_f)]$.
If $L_f$ is larger than 1, for large enough t, the term $(1-L_f^{t+1})/(1 - L_f)$ blows up and $C_0$ becomes $D/\epsilon_f$. Therefore, the upper bound of Lemma 1 becomes $D$. Here $D$ is the diameter of the state space, which is assumed to be bounded.

This carries to in the next lemmas. In Lemma 4, $C_5$ would be of the same order as $C_0$ (multiplied by an extra $L_1 L_f / (1 - \gamma) )$, so the upper bound of this lemma becomes proportional to D too.

The $C_0$'s appearance continues in the proof of Theorem 1, in which $C_8$ is proportional to $C_0$ and $C_5$. So, $C_8$ is also become proportional to $D/\epsilon_f$. When we have $C_8 \epsilon_f$ in Eq. (34), we get a constant term $D$.
A similar dependence appears in the proof of Theorem 2, where B_3 is proportional to $C_8 \epsilon_f$, which can be as large as $D$. And in Eq. (47), we have $B_3^2$. So the upper bound in Eq. (47), which seems to the be upper bound of Theorem 2, is proportional to $D^2$. This means that if $L_f$ is larger than one, the upper bound does not go to zero, no matter how small the model error $\epsilon_f$ is (unless it is actually zero). This makes the bound meaningless.

This might be unavoidable. I am not sure about it at the moment. But it definitely requires a discussion.

3) Assumption 2 has a term in the form of $E[\frac{s_{t_2}}{ s_{t_1}} ]$ (I have simplified the form). The states $s_{t_2}$ and $s_{t_1}$ are vectors in general. How is the division defined here?

4) Please improve the clarify of the proofs. For example, in Lemma 2 it seems that a negative sign is missing in Eq. (49). Also how do we get Eq. (50) and Eq. (52)? (I couldn't easily verify them).

5) I believe the "periodicity property" used in Assumption 1 should be "ergodicity property".

6) The paper still has a lot of typos, e.g., "To optimize the objective, One can ..." (P3), "argument data" (instead of augmented) (p4), "Superpose" (p5), "funcrion" (p6).

---

> ### Public Comment · ~Chongchong_Li1 · 2022-03-14
> **Updates for the Camera Ready Revision**
>
> We thank the AC for the comments on our work and here are updates for the revision of the paper.
>
> **1.	What happened to the noise term. Is it the error of the expected gradient of the next state**
> -	Yes, it is the error of the expected gradient in Eq5, and $\epsilon_f$ is defined in expectation. We have revised it in the updated version.
>
> **2.	The upper bound in Theorem 1**
> -	The consideration is reasonable, our theorem is fit for the case that the Lipschitz constant $L_f < 1$. For the dynamics that the gradient is always larger than 1, it is unrealistic and uncontrollable. For the dynamics that the gradient is always less than 1, theorem 1 and the assumptions are enough. For the last case, it needs a stronger assumption directly on $\Vert \tilde{s}_t-s_t\Vert$. We have added a remark in the updated version.
>
> **3.	How is the division defined in Assumption 2**
> -	$|| \mathbb{E}\left(\frac{s_{t_2}^1-s_{t_2}^0 }{s_{t_1}^1 - s_{t_1}^0 }\right) ||$ is the 2-norm of the matrix of which the element $ij$ is: $\mathbb{E}\left(\frac{s_{t_2}^1(i)-s_{t_2}^0(i) }{s_{t_1}^1(j) - s_{t_1}^0(j) }\right)$, where $s_{t}(i)$ is the i-th element in the vector $s_t$.
>
> **4.	Improve the clarity of the proofs**
> -	We added some details in the revised version.
>
> **5.	Ergodicity property**
> -	It is a typo and we revised it.
>
> **6.	Typos**
> -	Thank the reviewer for pointing out the typos, we have addressed them in the updated version.